# Rank Suggestion in Non-negative Matrix Factorization: Residual Sensitivity to Initial Conditions (RSIC)

**Marc A. Tunnell**                                          *tunnellm@mail.gvsu.edu*
*Grand Valley State University*

**Zachary J. DeBruine**                                      *debruinz@gvsu.edu*
*Grand Valley State University*

**Erin Carrier**                                            *carrieer@gvsu.edu*
*Grand Valley State University*

**Reviewed on OpenReview:** *https://openreview.net/forum?id=9Xj5w4DX0t*

## Abstract

Determining the appropriate rank in Non-negative Matrix Factorization (NMF) is a critical challenge that often requires extensive parameter tuning and domain-specific knowledge. Traditional methods for rank determination focus on identifying a single optimal rank, which may not capture the complex structure inherent in real-world datasets. In this study, we introduce a novel approach called Residual Sensitivity to Initial Conditions (RSIC) that suggests potentially multiple ranks of interest by analyzing the sensitivity of the relative residuals (e.g., relative reconstruction error) to different initializations. By computing the Mean Coordinatewise Interquartile Range (MCI) of the residuals across multiple random initializations, our method identifies regions where the NMF solutions are less sensitive to initial conditions and potentially more meaningful. We evaluate RSIC on a diverse set of datasets, including single-cell gene expression data, image data, and text data, and compare it against current state-of-the-art rank determination methods. Our experiments demonstrate that RSIC effectively identifies relevant ranks consistent with the underlying structure of the data, outperforming traditional methods in scenarios where they are computationally infeasible or less accurate. This approach provides a more scalable and generalizable solution for rank determination in NMF that does not rely on domain-specific knowledge or assumptions.

## 1 Introduction

Low-dimensional models of high-dimensional data are foundational for exploratory data analyses. Non-negative Matrix Factorization (NMF) has emerged as one such tool for data decomposition and analysis in various domains, including image processing (Guillamet et al., 2002; Lee & Seung, 1999; Liu et al., 2012), text mining (Hassani et al., 2019; Pauca et al., 2004), and bioinformatics (Devarajan, 2008; Gaujoux & Seoighe, 2010). By decomposing a non-negative matrix into non-negative factors, NMF is often able to extract meaningful patterns and components from otherwise hard-to-interpret datasets (Gillis, 2020). However, a critical challenge in applying NMF is determining the appropriate rank of decomposition (Wang & Zhang, 2013), which essentially dictates the number of components to extract from the data.

Traditionally, rank determination methods in NMF have largely focused on identifying a single "optimal" rank using heuristic methods or by leveraging additional knowledge of the distribution of the input data. While often useful, these methods have considerable limitations. They often require arbitrary parameter choices, are sensitive to the initialization, or depend on domain-specific knowledge, which may not always be available or easily interpretable. In this study, we introduce a novel approach to rank determination that seeks to suggest a number of ranks of interest instead of a single optimal rank.

Multiple ranks can be beneficial as, in most cases, there is no single one rank that matters (as NMF is fundamentally not a classification algorithm). Typically, the data is simply too large to examine in its raw form, and domain experts are looking for more usable lower rank representations from which they can glean insights. For instance, in bioinformatics, single-cell datasets are large, containing gene expression information. Bioinformatics researchers may glean useful insights from any number of different ranks, but cannot manually look through the full data or all ranks.

Our method, Residual Sensitivity to Initial Conditions (RSIC), is based on the observation that the reconstruction error of NMF is highly sensitive to the initial conditions of the factorization. This approach is grounded in a multi-resolution perspective by considering the stability of a rank's residual to its initial conditions. By doing so, we look to open up new avenues for interpreting NMF results, especially on complex datasets where a single rank may not be able to capture all relevant information.

Our method is designed to be general and applicable to a wide range of datasets, without requiring domain-specific knowledge or assumptions. Other methods, such as consensus-matrix methods (Brunet et al., 2004; Kim & Park, 2007), self-comparison methods (Hubert & Arabie, 1985; Grossberger et al., 2018; Sotiras et al., 2017), and cross-validation-based approaches (Muzzarelli et al., 2019; Owen & Perry, 2009; Kanagal & Sindhwani, 2010; Gilad et al., 2020), have been proposed in the literature to determine the rank of NMF. The vast majority of rank selection techniques have a strong preference for lower ranks, which may not always be appropriate for the data at hand. Our method on the other hand, does not have an algorithmic bias for lower ranks and is designed to suggest multiple ranks. Our methodology is applicable across a wide range of domains and does not rely on domain-specific parameters or *a priori* assumptions about the distribution of the data.

This paper is organized as follows. Relevant background information on NMF and the methods against which we compare are in Section 2. A detailed description of our method is given in Section 3. A high-level description of the datasets we compare on is given in Section 4. Our experimental setup is given in Section 5. The results are given in Section 6. Finally, a discussion and conclusion is given in Section 7.

## 2 Background

### 2.1 Non-negative Matrix Factorization

NMF is a low-rank matrix decomposition of an $m \times n$ non-negative matrix, $\boldsymbol{A}$, in which non-negativity constraints are imposed in computation of the lower rank factor matrices. Given a rank $k$ decomposition, the factor matrices $\boldsymbol{W}$ and $\boldsymbol{H}$ are $m \times k$ and $k \times n$ in dimension, respectively. Although a variety of methods for solving for $\boldsymbol{W}$ and $\boldsymbol{H}$ exist, such as hierarchical alternating least squares (Kimura et al., 2015; Gillis & Glineur, 2011) or gradient descent (Lee & Seung, 2000), we use Sequential Coordinate Descent (SCD) (Franc et al., 2005; Lin, 2007; Hsieh & Dhillon, 2011) and Multiplicative Update (MU) (Lee & Seung, 2000; Lin, 2007) exclusively in this study. Additionally, although a variety of objective functions exist, to allow for easier comparison with other work in this field, we minimize the Euclidean distance between the vectorized representation of $\boldsymbol{A}$ and the vectorized reconstruction, formulated as

$$\frac{1}{2} \left\| \boldsymbol{A} - \boldsymbol{W} \boldsymbol{H} \right\|_F^2, \tag{1}$$

where $\left\| \cdot \right\|_F$ denotes the Frobenius norm of a matrix. Given this minimization problem, which is subject to non-negativity constraints, and letting $\boldsymbol{B}_W = \boldsymbol{A}^T \boldsymbol{W}$, $\boldsymbol{B}_H = \boldsymbol{A} \boldsymbol{H}^T$, $\boldsymbol{G}_W$ be the gram matrix of $\boldsymbol{W}$, and $\boldsymbol{G}_H$ be the gram matrix of $\boldsymbol{H}^T$, the SCD update rules can be written in vector form as

$$\boldsymbol{H}_{i,:} \longleftarrow \max\left( 0, \boldsymbol{H}_{i,:} + \frac{(\boldsymbol{B}_W)_{i,:} - (\boldsymbol{H}^T \boldsymbol{G}_W)_{i,:}}{(\boldsymbol{G}_W)_{i,i}} \right),$$

for all $i \in \{1, 2, \ldots, k\}$, and

$$\boldsymbol{W}_{:,j} \longleftarrow \max\left( 0, \boldsymbol{W}_{:,j} + \frac{(\boldsymbol{B}_H)_{:,j} - (\boldsymbol{W} \boldsymbol{G}_H)_{:,j}}{(\boldsymbol{G}_H)_{j,j}} \right),$$

for all $j \in \{1, 2, \ldots, k\}$ (Lin, 2007; Hsieh & Dhillon, 2011; Franc et al., 2005). Note that the max function is applied element-wise and it is assumed that $\boldsymbol{W}$ and $\boldsymbol{H}$ are appropriately initialized prior to computation. Similarly, the MU rules can be written as

$$\boldsymbol{H} \longleftarrow \boldsymbol{H} \odot \left( \boldsymbol{B}_W^T \oslash (\boldsymbol{G}_W \boldsymbol{H}) \right)$$
$$\boldsymbol{W} \longleftarrow \boldsymbol{W} \odot \left( \boldsymbol{B}_H \oslash (\boldsymbol{W} \boldsymbol{G}_H) \right),$$

where $\odot$ and $\oslash$ denote the Hadamard product and division, respectively (Lee & Seung, 2000; Lin, 2007).

## 2.2 Limitations of Non-negative Matrix Factorizations

While NMF has been shown to be useful due to its ability to extract meaningful information, it has short-comings. First, the NMF factorization is not unique, making the problem of computing the NMF ill-posed (Gillis, 2012). There are typically numerous equally good solutions (e.g. factorizations for which the Frobenius norms of the residuals are equally small or big). Second, the underlying optimization problem is generally non-convex, meaning there is no guarantee that the obtained solution is a global minimum (e.g., there is no guarantee that the obtained factorization is the best factorization).

Additionally, it is well understood that NMF is highly sensitive to the initial conditions of the problem, and considerable work has gone into getting around this problem, though it is largely domain-dependent (Rosales et al., 2016; Yang et al., 2021; Devarajan, 2008). Furthermore, as discussed in Langville et al. (2014), many NMF initializations, such as the random, centroid, SVD-centroid, etc. exist, all with tradeoffs between performance (both convergence and quality of converged to solution) and computational complexity. Some initializations, such as centroid and SVD-centroid are very costly or rely on having other information from other common, yet expensive, factorizations. More recent work (Esposito, 2021) has looked comprehensively at a wider range of NMF initialization schemes, grouping them into different categories including random-based, deterministic low-rank approaches, clustering-based approaches, and evolutionary-based approaches. Choice of initialization is still an open question and there is no single accepted best practice.

Fundamentally, NMF is a low-rank decomposition, where the factorization can often be meaningfully interpreted as clustering. Under the view of NMF as an algorithm which produces a clustering, the matrix $\boldsymbol{W}$ assigns weights to basis vectors in $\boldsymbol{H}$, indicating how strongly each data point is associated with different clusters. These clusters, which are represented by the basis vectors of $\boldsymbol{H}$, often group similar data points together in a meaningful way. However, without additional constraints or sparsity enforcing conditions, the NMF is not a hard clustering algorithm (Kim & Park, 2008). Whether used for a soft clustering or not, in order for the factorization to be useful, it is crucial to know which rank decompositions are meaningful. Significant work has been done to this effect (as discussed in Section 2.4, with most studies focusing on classification datasets (for which class labels are known). Typically, this is to ensure that there is a "true rank". While this terminology is commonly used, we will instead be more specific and indicate "number of underlying classes". While NMF is often used for clustering (and it is not uncommon to evaluate clustering with classification data), fundamentally the decomposition of basis vectors, even at the same rank as the number of classes, may have no correlation with the classes considered as the true classes and may be meaningless.

## 2.3 Multi-Rank and Hierarchical Approaches to NMF

In many applications, the notion of a single "best" or "optimal" rank is complicated by the multi-resolution structure that is often embedded in real datasets. This is particularly relevant in single-cell transcriptomics, where an abundance of subtle subpopulations yield nested or overlapping signatures of gene coexpression (Kiselev et al., 2019; Wu & Wu, 2020). A single fixed rank may fail to capture these hierarchies, such as Tasic et al. (2016), which advocates for hierarchical decompositions instead of a single static factorization in their transcriptomics analysis. In one notable attempt to resolve complex hierarchies of cell state, Schwartz et al. (2020) employed divisive clustering to resolve cell type identity at multiple resolutions, showing recovery of rare cell types that at usual resolutions would have been unresolvable from their most similar common homologs.

In Brunet et al. (2004), the NMF concept of "metagenes" was introduced to interpret correlated features. While this was often done at a single rank, they recongnized that these "metagenes" can exist at multiple levels of granularity, ranging from broad cell-type signatures to fine-grained subclusters. Later studies in single-cell analysis highlight the inadequancy of a single rank for understanding cell mixtures (Kiselev et al., 2019; Wu & Wu, 2020). For example, Wu & Wu (2020) argue that discrete cell labels only approximate a true hierarchical topology, so a low-rank factorization can obscure additional relevant substructures that exist in the underlying data. Similarly, Tasic et al. (2016) utilized iterative clustering to reveal finer subpopulations in single-cell data, which parallels a multi-rank factorization approach in NMF.

Methodologically, multi-rank strategies begin by scanning over a range of candidate ranks (e.g., from 2 to some upper limit) and performing repeated factorizations. One then evaluates each solution for consistency, such as by measuring how often a similar factorization appears under different initial conditions or how stable the factors remain when the data are subsampled. Common metrics include silhouette scores, dispersion of the factor loadings, or residual reconstruction error. For example, Brunet et al. (2004) computes consensus matrices for each rank across multiple runs, and a rank is judged superior if its consesnsus matrix exhibits a clear block structure, which indicates consistent and well-separated factors.

Because no single criterion universally captures every nuance of a given dataset, many authors combine multiple diagnostics, such as reconstruction error, stability, and domain-specific interpretability, to arrive at a suitable rank. Kim & Park (2007) show that even small changes in a rank can drastically affect the factorization's interpretability, and they advocate for selecting a rank that yields a balance between parsimony and capturing sufficient structure of the underlying data. Meanwhile, Gilad et al. (2020) emphasizes that no single rank determination strategy can be trusted across all datasets. Certain rank determination methods are biased towards low-rank representations of data while other methods may reveal a rank that is not particularly relevant to domain scientists.

In practice, this means that domain knowledge is essential when determining whether useful insights can be drawn from a given rank. In the case of single-cell transcriptomics, biologically meaningful metrics, such as whether factors align with known pathways or cell types, may override purely statistical considerations. Consequently, multi-rank scanning, combined with robust evaluation criteria and domain-based validation, offers a more comprehensive and useful view than forcing the data into a single, potentially under- or over-factored representation.

## 2.4 Previous Work on Rank Determination

A variety of methods to determine the rank have been proposed in the literature. These can largely be broken into three main categories: consensus-matrix methods, self-comparison methods, and cross-validation-based approaches. A brief overview for each of these methods is provided in this section and further details for the implementations we use for all methods we compare against are given in Section 5.

### 2.4.1 Consensus-Matrix Methods

Cophenetic correlation and dispersion coefficient both compute a consensus matrix based on the clustering obtained with NMF, then compute their metric based on this matrix (Kim & Park, 2007; Brunet et al., 2004). In both cases, the consensus matrix is computed and then averaged over a number of random starts.

After computing the averaged consensus matrix, both methods compute their metric. Cophenetic correlation then chooses a rank based on when the cophenetic correlation first begins to drop (Brunet et al., 2004). However, it is important to note that what constitutes a drop is dependent on how many ranks are plotted as this affects the range of the $y$-axis and consequently what appears to be a drop. Alternatively, the dispersion coefficient method chooses a rank based on when the dispersion coefficient is maximized (Kim & Park, 2007).

In the literature, both methods are often run on a relatively small range of ranks around the point at which the authors expect the optimal rank to be. With cophenetic correlation, the optimal rank is chosen based on an extremely small drop in the coefficient over the range. In fact, one run in Brunet et al. (2004) was shown to be inconclusive based on the lack of a drop on the short range of ranks tested. In our testing,

we find the behavior of cophenetic correlation to be erratic on all of our datasets when testing higher ranks than was tested in the original implementation. Similarly, we find the dispersion coefficient to be generally increasing as a function of rank after the initial drop. In both cases, these metrics leave the user unsure of what to pick unless choosing to run on a relatively small few ranks. It is not always possible to determine an appropriately small range to check, especially when the rank of the underlying data cannot be determined *a priori*. Furthermore, the number of ranks included when plotting the cophenetic coefficient changes how stretched the plot is left-to-right between consecutive ranks, which drastically affects how steep a drop appears to be.

### 2.4.2 Self-Comparison Methods

Self-comparison methods can be subset into two categories, split validation and permutation comparison. Split validation cuts the input matrix randomly in half, reorders the halves by the similarity of the basis vectors, then computes the similarity between the two halves. This similarity metric can take a few forms but has shown success in the past using Adjusted Rand Index (ARI) (Hubert & Arabie, 1985; Grossberger et al., 2018) and inner product (Sotiras et al., 2017). We choose not to compare against inner product as it had poor performance on real data in the testing performed by Muzzarelli et al. (2019).

Permutation compares the slope of the elbow of the reconstruction error of the factorization of $A$ against the slope of the elbow of the reconstruction error of a permuted version of $A$. Effectively, this compares the ability of NMF to reconstruct the dataset against the ability of NMF to reconstruct a random matrix of exactly the same magnitude as the original matrix. Effectively, when the slope of the reconstruction error of $A$ is equal to that or greater than the slope of the permuted matrix, no extra information is able to be extracted from the original dataset compared to a random one.

### 2.4.3 Non-Categorized Methods

The elbow method is a popular technique for rank determination used in cluster analysis. This method involves plotting the residual as a function of $k$ and picking the elbow of the curve as the correct number of clusters to use. The elbow in the graph is where the rate of decrease changes, representing the point at which increasing the number of clusters does not significantly improve the fit of the model. Although the elbow method is at least partially subjective, we compare against it for completeness.

Akaike Information Criterion (AIC) was successful in determining rank on time-series data in Cheung et al. (2015) using a modified implementation of NMF. We choose not to compare against AIC as it had poor performance in Gilad et al. (2020) and was otherwise criticized by Ito et al. (2016) for requiring assumptions that do not necessarily hold in NMF. In this study, we focus on a general approach that does not make statistical assumptions regarding NMF or the underlying data.

Similarly, we do not compare against Cai et al. (2022), a sequential hypothesis testing method, due to their requirement that the underlying data follows certain distributions.

For similar reasons, we do not compare against the category of Bayesian methods due to their requirements for *a priori* knowledge of the distribution for prior estimation (Schmidt et al., 2009; Cemgil, 2009).

Additionally, we do not compare against methods utilizing minimum description length (MDL) as they assume a statistical model of the NMF (Yamauchi et al., 2012).

Relevance determination is worthy of mention here but is ultimately not relevant to the discussion in this paper. Essentially, these methods identify relevant clusters given a larger rank NMF decomposition (Tan & Févotte, 2009).

### 2.4.4 Cross-Validation-Based Approaches

A variety of methods for NMF rank determination using cross-validation (CV) have been proposed in the literature. These include bi-CV by Owen & Perry (2009), and imputation-based CV by Kanagal & Sindhwani (2010). We choose not to compare against bi-CV as the results have been shown in practice to be unclear or unstable by Kanagal & Sindhwani (2010); Gilad et al. (2020).

Imputation-based CV is performed by optimizing for $\boldsymbol{W}$ and $\boldsymbol{H}$ given an imputed version of $\boldsymbol{A}$ in which a percentage of values are denoted as missing (Kanagal & Sindhwani, 2010). The use of a Wold holdout pattern has been shown to be performant and is most widely used (Wold, 1978; Kanagal & Sindhwani, 2010). We denote withheld values as 1 in the binary masking matrix, $\boldsymbol{M}$, and 0 otherwise. These values are hidden from computation during optimization. In most implementations, the reconstruction error is afterward calculated as

$$\frac{\|\boldsymbol{M} \odot (\boldsymbol{A} - \boldsymbol{W}\boldsymbol{H})\|_F^2}{\|\boldsymbol{M}\|_F^2}. \tag{2}$$

In the implementation originally proposed by Kanagal & Sindhwani (2010), this is performed multiple times per rank and then averaged; the rank with lowest mean reconstruction error is chosen. For the purpose of comparison in this study, we will call this method KS-CV, based on the last initials of the authors.

A variety of methods have been proposed that directly build on the work of Kanagal & Sindhwani (2010) such as MADImput (Muzzarelli et al., 2019), MSEImput (Muzzarelli et al., 2019), and CV2K (Gilad et al., 2020). Each of these three methods optimizes over every rank of interest a number of times, as was performed by Kanagal & Sindhwani (2010). For MADImput, the Median Absolute Deviation (MAD) of the reconstruction errors is calculated at each rank and the rank with lowest MAD is chosen (Muzzarelli et al., 2019). We choose not to compare against MSEImput because it performed poorly on all but simulated data (Muzzarelli et al., 2019). Differing from the other imputation methods described, the authors of Gilad et al. (2020) compute the error as the $L^1$-norm of the error over the masked values, computed against a normalized version of the initial matrix which allows them to normalize both $\boldsymbol{W}$ and $\boldsymbol{H}$, as detailed in Algorithm 2 in their paper. The rank with minimum median reconstruction error calculated as stated is chosen, but is adjusted down based on a correction step determined by a Wilcoxen rank-sum test (Gilad et al., 2020).

## 2.5 Complexity

The majority of rank determination methods compute NMF using a standard optimization algorithm a number of times, then compute their metric. While the cost of computing these metrics is not free, it is typically substantially less than the cost of computing the NMF decomposition. On the other hand, imputation-based methods must deal with missing values during the computation of the NMF decomposition, fundamentally altering how the NMF decomposition is computed. The complexity of imputation-based CV methods is often overlooked but is a significant burden in practice.

Before discussing the complexity of computing an NMF decomposition with missing values, we must first discuss the complexity of a standard NMF decomposition. As is common practice, we only count multiplication and division, and we assume naive algorithms for common operations such as matrix multiplication. The amount of work performed at a given rank for a single optimization iteration using SCD when no values are missing is described by

$$2mnk + 2mk^2 + 2nk^2. \tag{3}$$

This assumes two Gram matrix computations, one computation of $\boldsymbol{B}_W$ and $\boldsymbol{B}_H$, and one computation of $\boldsymbol{W}\boldsymbol{G}_H$ and $\boldsymbol{H}^T\boldsymbol{G}_W$. Under the assumption that $k \ll \min(m,n)$, the time complexity is $\mathcal{O}(mnk)$. The amount of work required for a single iteration of NMF with MU is slightly more, but results in the same overall time complexity.

The computation of CV with missing values is considerably more complex. The implementation provided by Lin & Boutros (2020), and used by Gilad et al. (2020), implements imputation-based CV by creating a new Gram matrix for each row or column affected by missing values. That is to say that each column, $\boldsymbol{H}_{:,j}$, in the update of $\boldsymbol{H}$ requires a different Gram matrix, denoted by $\boldsymbol{G}_W^{(j)}$, and created as

$$\begin{aligned}
\boldsymbol{G}_W^{(j)} &= (\operatorname{diag}(\sim\boldsymbol{M}_{:,j})\boldsymbol{W})^T (\operatorname{diag}(\sim\boldsymbol{M}_{:,j})\boldsymbol{W}) \\
&= \boldsymbol{W}^T\operatorname{diag}(\sim\boldsymbol{M}_{:,j})\boldsymbol{W},
\end{aligned}$$

where $\sim$ denotes element-wise logical negation. Similarly, each row, $\boldsymbol{W}_{i,:}$, in the update of $\boldsymbol{W}$ requires a different Gram matrix, $\boldsymbol{G}_H^{(i)}$, which is created as

$$\begin{aligned} \boldsymbol{G}_H^{(i)} &= \left(\boldsymbol{H}\mathrm{diag}(\sim\boldsymbol{M}_{i,:})\right)\left(\boldsymbol{H}\mathrm{diag}(\sim\boldsymbol{M}_{i,:})\right)^T \\ &= \boldsymbol{H}\mathrm{diag}(\sim\boldsymbol{M}_{i,:})\boldsymbol{H}^T. \end{aligned}$$

Taking into account the need to compute all of these Gram matrices, the missing value computation cost per iteration is captured by

$$2mnk^2 + 2mnk + mk^2 + nk^2. \tag{4}$$

This assumes the computation of $m + n$ Gram matrices, one computation of $\boldsymbol{B}_W$ and $\boldsymbol{B}_H$, one computation of $\boldsymbol{W}_{i,:}\boldsymbol{G}_H^{(i)}$ for all $i \in \{1, 2, \ldots, m\}$ and $(\boldsymbol{H}_{:,j})^T\boldsymbol{G}_W^{(j)}$ for all $j \in \{1, 2, \ldots, n\}$, and at least one missing value per row and column. This gives a time complexity of $\mathcal{O}(mnk^2)$, but attention should be given to the two equations which model their behavior. The dominating term of Equation 3 is contained within Equation 4, meaning that nearly all of the work required to compute the original factorization must be performed in addition to the additional gram matrix related work. For datasets of relatively small size, this is not particularly burdensome, but it is entirely unfeasible on larger datasets as discussed later in Section 6.3. It should be noted that the CV2K implementation from Gilad et al. (2020) has lower time complexity but necessitates the in-memory storage of five additional matrices of equal size to $\boldsymbol{A}$; this is similarly burdensome for matrices of sufficient size.

In practice, as the computed NMF decomposition is dependent on the initialization, essentially all rank detection methods compute the NMF decomposition for many different initializations. Let $a$ be the desired number of initializations for each rank. Let $t$ be the number of iterations. Then, the overall standard NMF decomposition cost for a given rank $k$ is

$$at(2mnk + 2mk^2 + 2nk^2). \tag{5}$$

In contrast, the total CV-based NMF decomposition cost with missing values for a given rank $k$ is

$$at(2mnk^2 + 2mnk + 2mk^2 + 2nk^2).$$

## 3 Residual Sensitivity to Initial Conditions (RSIC)

As previously described in Section 2.2, NMF is highly sensitive to the initial conditions of the $\boldsymbol{W}$ and $\boldsymbol{H}$ matrices. While many initializations exist, as discussed in Section 2.2, we limit our scope to random initializations in this work. We have found that, even amongst factorizations whose residuals have comparable Frobenius norms, the reconstruction error computed at any point may be arbitrarily worse or better in any given factorization. This observation highlights the inherent unpredictability in NMF due to random initializations and can be seen clearly in Figure 1. This figure shows the delta between the smallest and largest relative reconstruction error at each point in the rank-10 reconstruction of the Faces dataset (described in detail later in Section 4.2.3). Note that the relative reconstruction error for a point $(i, j)$ is computed as

$$\left| \frac{a_{i,j} - \boldsymbol{w}_{i,:}\boldsymbol{h}_{j,:}}{1 + a_{i,j}} \right|.$$

The maximum delta figure is plotted over the models which had within 10% Frobenius norm of the residual of the median residual model. This figure shows that, even within models that have comparable norms, there exists massive deviation in the reconstruction error when considered coordinatewise. Similar behavior was found to be present in all of the datasets that were tested.

Of interest, these observations reveal a pattern within the variability: at certain ranks, the sensitivity to initial conditions as measured by the deltas in reconstruction error diminishes greatly, forming what can be described as "islands of stability". These ranks, where the variance in reconstruction error is minimal despite different initializations, stand out against nearby ranks that exhibit high sensitivity.

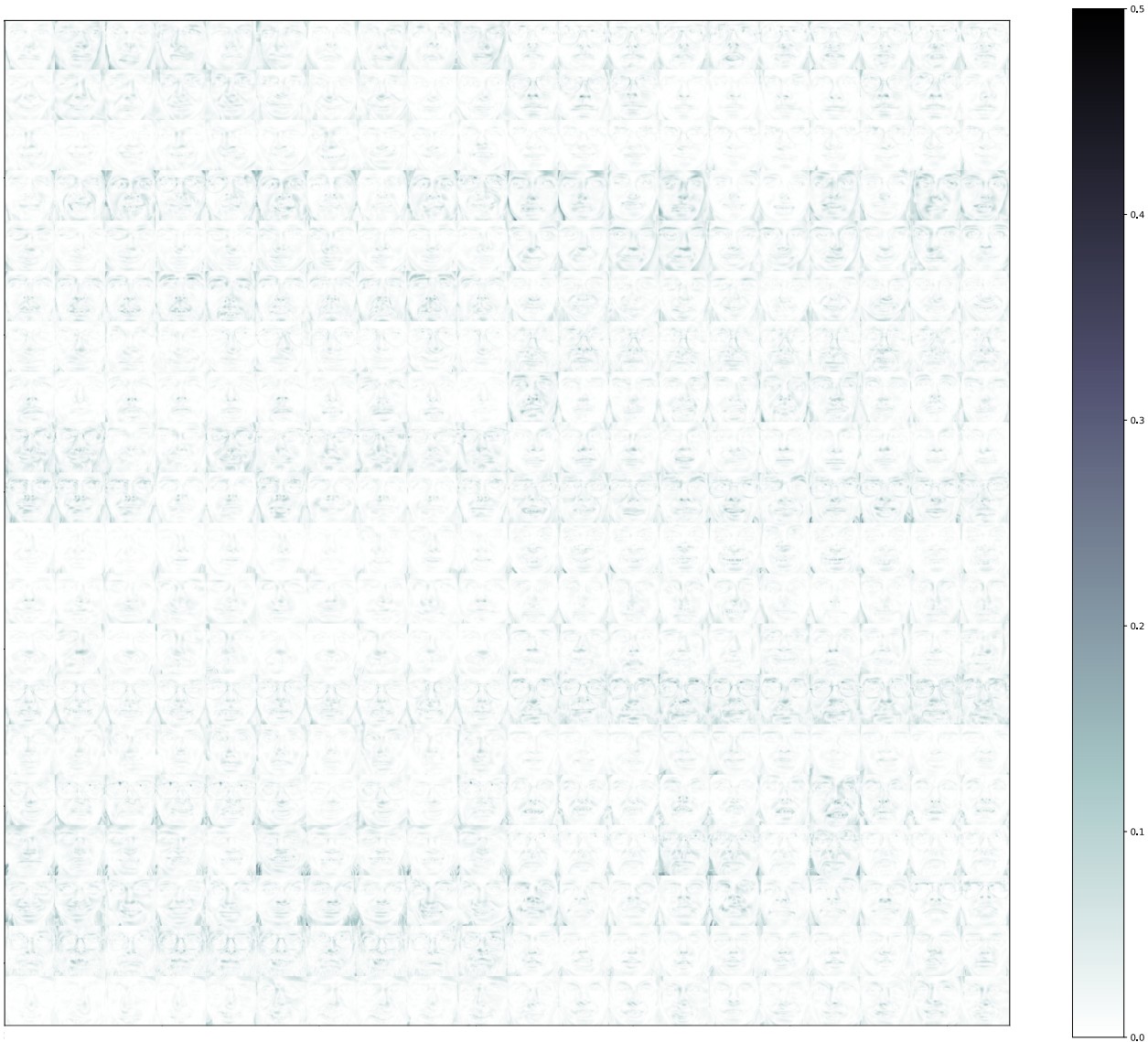

Figure 1: The delta between the smallest and largest relative reconstruction error at each point in the rank-10 reconstruction of the 400 faces in the Faces dataset.

We hypothesize that these "islands of stability" are not merely random occurrences but could be indicative of inherent structure or patterns within the data that are particularly well-captured by NMF at these specific ranks. These stable ranks may correspond with factorizations that more effectively distill the essential features of the data and are less influenced by the initial conditions.

To investigate this, we compute a number of factorizations at each rank, where the randomly generated initial conditions are related across ranks in a scheme described next in Section 3.1. We then attempt to quantify the Residual Sensitivity to Initial Conditions (RSIC) at a given rank using the metric developed in Section 3.2. Plotted as a function of the rank, $k$, we then identify "islands of stability" as potential ranks of interest that should be further investigated. Through general testing, we identify 100 as the approximate number of random initializations needed for consistent behavior regardless of seed, with more providing no meaningful benefit.

### 3.1 Progressive Random Initialization

We implement a progressive random initialization scheme, which allows individual random initializations to be related across ranks, smoothing out variations in reconstruction error across ranks of the NMF factorization for a single progressive random initialization. For a given initialization with maximum rank of interest, $k_{\max}$, let $\mathbf{W}_{\text{init}}$ and $\mathbf{H}_{\text{init}}$ be random matrices of dimension $m \times k_{\max}$ and $k_{\max} \times n$, respectively, whose entries are generated from the uniform distribution on the half interval $[0, 1.0)$. Assume $a$ is the desired number of initializations per rank. Let $\mathbf{W}_{\text{init}}^{(r)}$ be the $r$-th random initialization of $\mathbf{W}_{\text{init}}$, and $\mathbf{W}_{\text{init}}^{(r,k)}$ is a copy of the left submatrix $\left(\mathbf{W}_{\text{init}}^{(r)}\right)_{:,:k}$.

Likewise for $\mathbf{H}_{\text{init}}$, let $\mathbf{H}_{\text{init}}^{(r,k)}$ be a copy of the upper submatrix $\left(\mathbf{H}_{\text{init}}^{(r)}\right)_{:k,:}$. Then, for a given initialization, $r$, and letting $k_{\min}$ be the minimum rank of interest, we have the ordered set of tuples of initial matrices,

$$\mathbb{S}_{\text{init}}^{(r)} = \left\{ \left(\mathbf{W}_{\text{init}}^{(r,k_{\min})}, \mathbf{H}_{\text{init}}^{(r,k_{\min})}\right), \left(\mathbf{W}_{\text{init}}^{(r,k_{\min}+1)}, \mathbf{H}_{\text{init}}^{(r,k_{\min}+1)}\right), \ldots, \left(\mathbf{W}_{\text{init}}^{(r,k_{\max})}, \mathbf{H}_{\text{init}}^{(r,k_{\max})}\right) \right\}.$$

Assume a set, $\mathbb{S}_{\text{init}}^{(*)}$, exists for each progressive random initialization we are performing. For simplicity, we index these sets as

$$\mathbb{S}_{\text{init}}^{(r,k)} = \left(\mathbf{W}_{\text{init}}^{(r,k)}, \mathbf{H}_{\text{init}}^{(r,k)}\right).$$

Given this indexing, this specifically means that the matrices in $\mathbb{S}_{\text{init}}^{(*,k-1)}$ contain a copy of the respective matrices in $\mathbb{S}_{\text{init}}^{(*,k)}$ for all $k \in \{k_{\min}+1, k_{\min}+2, \ldots, k_{\max}\}$. This is such that $\mathbf{W}_{\text{init}}^{(*,k-1)}$ contains a copy of the left submatrices of all but the last column of $\mathbf{W}_{\text{init}}^{(*,k)}$ and $\mathbf{H}_{\text{init}}^{(*,k-1)}$ contains a copy of the upper submatrices of all but the last row of $\mathbf{H}_{\text{init}}^{(*,k)}$.

Finally, let

$$\mathbb{S}_{\text{nmf}}^{(r,k)} \longleftarrow \text{NMF}\left(\boldsymbol{A}, \mathbb{S}_{\text{init}}^{(r,k)}\right)$$

be the resulting factorization of $\boldsymbol{A}$ after performing NMF with a given rank and initialization pairing. This factorization transforms the unoptimized matrices, giving their optimized counterparts. This is performed over all progressive random initializations and ranks, giving $\mathbb{S}_{\text{nmf}}$.

### 3.2 Mean Coordinatewise Interquartile Range (MCI)

The goal of this metric is to capture the sensitivity of the relative reconstruction error at a given rank by measuring the average spread of relative reconstruction errors at each point. This is computed by taking the interquartile range (IQR) of the relative reconstruction error at each point over the number of initializations.

Let $\boldsymbol{R}^{(k)}$ be a matrix of dimension $a \times mn$, and assume $\text{vec}(\cdot)$ flattens an $m \times n$ matrix to a $mn$ dimension vector in any consistent ordering. The values of $\boldsymbol{R}^{(k)}$ are given by

$$\boldsymbol{R}_{r,:}^{(k)} = \text{vec}\left(\boldsymbol{A} - \boldsymbol{W}_{\text{nmf}}^{(r,k)} \boldsymbol{H}_{\text{nmf}}^{(r,k)}\right) \oslash \left(\boldsymbol{e} + \text{vec}(\boldsymbol{A})\right),$$

for all $r \in \{1, 2, \ldots, a\}$ with given rank, $k$, and where $\boldsymbol{e}$ is the vector of ones. The Mean Coordinatewise IQR (MCI) at each rank, $k$, is computed as

$$\text{MCI}^{(k)} = \frac{1}{mn} \sum_{j=1}^{mn} \text{IQR}\left(\boldsymbol{R}_{:,j}^{(k)}\right),$$

where IQR computes the interquartile range of a vector. This is performed for all $k \in \{k_{\min}, k_{\min}+1, \ldots, k_{\max}\}$.

Computing the MCI over a large number of factorizations requires the storage of all of these factorizations. Although this may be a significant amount of space for large matrices, the total amount of work required is

significantly less than those methods based on cross-validation techniques. Additionally, these matrices may be stored on disk and do not need to be kept in memory.

In this study, we choose ranks that have a relatively low RSIC-MCI value compared to their neighbors of higher rank. To do this, we plot RSIC-MCI as a function of rank and look for ranks in pronounced dips or flat-lining after which the metric increases. To tie break along regions where multiple ranks exhibit the same flat behavior, we take the last rank prior to an increase in the metric, as that rank is equally as stable, but further decomposes the data. In practice, one may be interested in looking at each of these tied ranks.

### 3.3 Scalability and Work

The scalability of RSIC is a significant advantage over many existing methods. Like self-comparison methods, the cost of RSIC is dominated by the computational cost of computing the NMF decomposition. Once an NMF decomposition is computed, the cost of computing the MCI metric is dominated by the cost of computing the residual, which requires one matrix multiplication per initialization per rank. Thus, the total cost to compute MCI metric for a given rank $k$ is $\mathcal{O}(amnk)$, and the total cost to compute both the NMF decomposition and the metric for a given rank $k$ is

$$amnk + at\left(2mnk + 2mk^2 + 2nk^2\right).$$

Note that when compared to Equation 5, the added cost of computing the MCI is not the dominant cost, and the overall computational complexity for a given rank $k$ remains $O(atmnk)$. Thus, even including the computational cost of computing the MCI metric, it is still a substantial improvement over the cost for imputation-based methods, which require $O(atmnk^2)$ work for a given rank $k$.

**Storage Costs**  The primary burden of the RSIC method is the high storage requirements of storing the factorizations. This storage cost is given as

$$\mathcal{O}\left(k(m+n)\right)$$

double precision floating point numbers for a given rank $k$. More precisely, the overall storage cost for all ranks and initializations is given by

$$8\sum_{r=k_{\min}}^{k_{\max}} a\,r\,(m+n) \;=\; 8a\,(m+n)\,\frac{\left(k_{\max}-k_{\min}+1\right)\left(k_{\max}+k_{\min}\right)}{2}$$

bytes, again assuming double precision floating point numbers. For example, on a real-world dataset we later use: suppose $a = 100$ on a problem with dimensions $m = 11967$ and $n = 28095$. Setting $k_{\min} = 2$ and $k_{\max} = 64$, the storage cost is 66.63GB. As the size of the datasets increases, the storage cost will increase linearly with any given dimension, number of initializations, or maximum rank of interest. Of course, these matrices can be kept on disk and loaded into memory as needed in later steps of the computation.

**Parallelizability**  A major advantage of RSIC is that it is trivially parallelizable. Factorizations at different ranks and with different initializations are computed independently. Although there would appear to be a dependence due to the reliance on the progressive random initialization scheme, it is possible to produce these progressive initializations in any order. Thus, for the $a$ initializations and $k$ ranks, a parallel implementation of RSIC may be performed on $ak$ different machines. Further, the computation of the MCI metric is only partially dependent on the computation of initializations and may begin to be computed as soon as the first initialization is complete. For simplicity, we will assume that the computation of the MCI is started once all initializations have finished computation.

The computation of the MCI for each rank is independent from other ranks, so we focus only on the computation of the MCI for a single rank. For a given rank, the MCI can be computed in a batched manner, where the metric is computed for a batch of $b$ initializations at a time. These batches can be trivially split over whichever dimension of the original matrix is most convenient. We note that it is possible (and would potentially lead to better parallel performance) to split the computation of the MCI in a 3-dimensional manner over the rows, columns, and initializations, but we do not explore that in this paper.

For the trivial splitting along columns, and for the first batch of columns ranging from 1 to $b$, the computation of the residual is given by

$$\boldsymbol{R}_{r,1:mb}^{(k)} = \mathrm{vec}\left(\boldsymbol{A}_{:,1:b} - \boldsymbol{W}_{\mathrm{nmf}}^{(r,k)}\left(\boldsymbol{H}_{\mathrm{nmf}}^{(r,k)}\right)_{:,1:b}\right) \oslash \left(\boldsymbol{e} + \mathrm{vec}\left(\boldsymbol{A}_{:,1:b}\right)\right),$$

for all $r \in \{1, 2, \ldots, a\}$. Then computing the partial MCI on the first batch is given by

$$\mathrm{MCI}_{\mathrm{partial}}^{(k)} = \frac{1}{mn}\sum_{j=1}^{mb}\mathrm{IQR}\left(\boldsymbol{R}_{:,j}^{(k)}\right).$$

The second batch would be processed for $\boldsymbol{R}_{r,mb+1:2mb}^{(k)}$ and similar for computing the partial MCI, where each batch is completely independent of each other. To combine the partial MCI values, we have

$$\mathrm{MCI}^{(k)} = \sum \mathrm{MCI}_{\mathrm{partial}}^{(k)}$$

for all partial MCI for the given rank $k$. We note that, when splitting by columns, the vectorization of the matrix must be done column-wise and row-wise when splitting by row.

Thus the computation of the MCI is able to be performed in a parallel batch-wise manner. This adds no overhead in terms of work and requires no communication between threads except to reduce the final MCI value.

## 4 Datasets

As discussed in Section 2.2, there is no real concept of a true rank and we instead will indicate the number of underlying classes for each dataset for which it is known. Additionally, certain datasets do not have a single number of underlying classes as sub-classes may exist which equally make sense to target in the analysis of a dataset. We discuss any potential sub-classes in our discussion of each dataset. While presented for comparison with the literature, the number of underlying classes does not necessarily correlate directly, if at all, to the optimal rank of an NMF decomposition. We compare our method on eight datasets from three different disciplines. We break this section into three subsections, one for each type of dataset used. All datasets are formatted such that each row represents a sample and the columns within a row are features. This means that the two single cell datasets, introduced next, are transposed from how they are generally presented in the literature.

### 4.1 Single Cell Datasets

ALL-AML was originally described in Golub et al. (1999) and retrieved using the package provided by Gaujoux & Seoighe (2023). This dataset is $38 \times 5000$ in dimension, consisting of the 5000 most highly varying human genes in the original dataset and taken from 38 bone marrow samples (Golub et al., 1999; Gaujoux & Seoighe, 2023). Of these 38 samples, 27 are related to acute lymphoblastic leukemia (ALL) and 11 acute myeloid leukemia (AML) (Golub et al., 1999). Given the existence of two cancer types, the number of underlying classes is technically 2, though most authors find a rank 3 NMF decomposition to be more informative (Brunet et al., 2004).

The PBMC3K dataset was retrieved from 10x Genomics (2016) and has dimension $2700 \times 13714$. This dataset consists of 2700 cells and 13714 genes and comes pre-filtered for relevance. This dataset has no consensus factual number of underlying classes but has been shown using domain knowledge to contain 9 distinct cell types (Du et al., 2020), which will be treated as the number of underlying classes for our purposes.

### 4.2 Image-Based Datasets

### 4.2.1 Swimmer

The Swimmer dataset was first described in Donoho & Stodden (2003) and retrieved from the package released by Tan & Févotte (2009).[1] Swimmer has dimension $256 \times 1024$ and is a synthetic dataset consisting of 256 flattened $32 \times 32$ black and white stick figure images meant to mimic a variety of breaststroke positions. There are a total of 16 limb positions, giving a total number of underlying classes of 16.

### 4.2.2 Full Digits & Dig0246

We retrieved the "Optical Recognition of Handwritten Digits" dataset by Alpaydin & Kaynak (1998) from scikit-learn, a scientific computing package released by Pedregosa et al. (2011). We use this both as the full dataset and additionally use a selected subset for experimentation. We call the full dataset Full Digits and the subset Dig0246, containing only the digits $\{0, 2, 4, 6\}$. We subset in this manner for the sake of comparison and consistency with the authors of Muzzarelli et al. (2019), who did not run on the full dataset. We should note that the clustering behavior of NMF appears to more clearly separate the classes in Dig0246 than with Full Digits. An example of this clustering behavior will be shown later in Section 6.2. The number of underlying classes are 10 and 4 for Full Digits and Dig0246, respectively.

### 4.2.3 Faces

We retrieved the "AT&T Laboratories Cambridge Faces" dataset by Cambridge (1994) using the package provided by Pedregosa et al. (2011). This dataset has dimension $400 \times 4096$, consisting of 400 flattened black and white images of size $64 \times 64$. The dataset consists of 10 people, with 40 images from each. Given the number of subjects, the number of underlying classes is 10. The images were originally $92 \times 112$ in dimension but have been modified in the version retrieved from Pedregosa et al. (2011).

### 4.3 Text-Based Datasets

All text datasets were transformed using a bag-of-words model, `CountVectorizer`, provided by Pedregosa et al. (2011). The maximum document frequency was set to 95%, minimum document frequency to an integer value of 2, and the stop-words set to `english` as provided by the implementation. The maximum features was set to 4000 and 30000 for NewsGroup4000 and Web of Science, respectively. All other options were left to their defaults.

### 4.3.1 NewsGroup4000

We obtained the NewsGroup4000 dataset, originally found at Lang (1997), using the packaged provided by Pedregosa et al. (2011). This dataset is of dimension $11314 \times 4000$ and contains 20 topics ranging from baseball to medical. Many of the 20 topics belong to high-level topics including "comp", "rec", "science", "misc", "talk", "alt", and "soc". In some cases there are additional levels between the high-level topics and the low-level topics. Based on the number of topics, we consider number of underlying classes as 20, although due to the different levels of topics, there is no single number of underlying classes.

### 4.3.2 Web of Science

We obtained the Web of Science dataset, which was originally described in Kowsari et al. (2017) from Kowsari et al. (2019). The data was preprocessed as previously described using `CountVectorizer`. This dataset is of dimension $11967 \times 28095$ and contains 11967 documents from 35 categories and 7 parent categories. Fundamentally, there are two different accurate values for the number of underlying classes in this dataset, 35 and 7.

---

[1]The package may be found at `www.irit.fr/~Cedric.Fevotte/extras/pami13/ardnmf.zip`

# 5 Experimental Setup

We opt to use only publicly available packages for all significant computations. Due to the large variety of datasets and methods we compare, for computational feasibility, we perform 100 random initializations for each method on each dataset. For each dataset, we let $k_{\min} = 2$ and $k_{\max} = \min(m, n, 64)$ and perform each method on each rank between $k_{\min}$ and $k_{\max}$ inclusively. Since it has been shown that MU converges to a solution more slowly than SCD (Lin & Boutros, 2020), we use SCD as the optimization routine for all methods in which the option is available. We force all methods using SCD to run to 100 optimization iterations per initialization and rank by setting the tolerance to $1e-16$. For the remaining methods in which SCD is not an option, MU is selected and forced to run for 500 optimization iterations due to its slower convergence. For consistency with Muzzarelli et al. (2019), all methods optimize the Frobenius norm defined in Equation 1. As mentioned previously, we limit our comparison to a subset of metrics and initializations initializations for the reasons described in Section 2. Before the start of computation for each method and dataset, the random state is set to the seed 123456789. For the computation performed with the packages provided by Gilad et al. (2020) and Lin & Boutros (2020), which offer multi-threading support, computation is performed on a dedicated workstation with a 32 thread Threadripper processor and 64 GB of memory. All other computation is performed sequentially on a consumer-grade Intel processor with 16 threads and 128 GB of memory.

## 5.1 Elbow

For the comparison against the elbow method, we determine the elbow based on the average of the reconstruction error at each rank. The results are provided based on a visual determination of where the bend appears to be. We utilize an averaging here to smooth the plotted curve, which makes determining the elbow a reasonable task. In our testing, we find that there is little difference between using the mean or median.

## 5.2 Cophenetic Correlation & Dispersion Coefficient

For the comparison with cophenetic correlation and dispersion coefficient methods (Brunet et al., 2004; Kim & Park, 2007), we use the NIMFA package provided by Zitnik & Zupan (2012). This package does not have support for SCD, so we use the MU optimizer instead. The initialization type is set to `random`, the number of initializations to 100, and the number of iterations to 500. The update type is set to `euclidean` and the objective function to `fro`. We pass in the range of ranks from $k_{\min}$ to $k_{\max}$ and plot the cophenetic correlation coefficient and dispersion coefficient as determined by the package. We then choose the optimal rank for each based on the criteria set forth by the original authors in Brunet et al. (2004) and Kim & Park (2007) and briefly described in this paper in Section 2.4.1.

## 5.3 Permutation

For the comparison with permutation, we perform the optimization of NMF using the implementation provided by Pedregosa et al. (2011). We compute the Frobenius norm of both the permuted and non-permuted matrix as a function of rank. Using SciPy's implementation of the Savitzky-Golay filter (Virtanen et al., 2020), we approximate the slope of the residuals as a function of rank. We look for the point where the reconstruction error of the unpermuted matrix decreases less sharply than that of the permuted matrix. This corresponds to the point at which the slope of the residuals of the non-permuted matrix is greater than or equal that of the slope of the residuals of the permuted matrix because both are decreasing and the slopes are negative. In terms of the approximated derivative, the rank selected is the point immediately before overtaking the elbow of the permuted matrix, or the point at which they are exactly equal. Due to floating point arithmetic, equality is considered a relative tolerance of $1 \times 10^{-8}$, computed relative to the larger of two derivative estimates. This is performed 100 times and the median result is computed.

### 5.4 ARI Method

For the comparison against the ARI method, we randomly split the matrix into two equal parts along the dimension relating to factors. If splitting into even parts is not possible, the larger split is truncated to size. Then NMF is computed on each split using the implementation provided by Pedregosa et al. (2011). The cosine distance between factors is computed using the package provided by Virtanen et al. (2020), giving a distance matrix. This is then passed to the `linear_sum_assignment` function as provided by Virtanen et al. (2020), which implements the Hungarian algorithm described in Crouse (2016). The result is the ARI, which is then averaged over all initializations at a given rank. The rank with highest mean ARI is selected. We note that we ran with both the mean and median ARI, which resulted in the same decided rank in all cases.

### 5.5 Cross Validation Methods

For the comparison against KS-CV and MADImput, we use the package provided by Lin & Boutros (2020). The output of this package is the reconstruction error at each rank as previously defined in Equation 2. Using this output, we are able to compute each of these metrics and choose a rank based on where it is minimized. For the comparison against CV2K, we use the package provided by Gilad et al. (2020), which returns both a similar output and the chosen rank based on their criteria.

We note that we have chosen to use a different initialization scheme than was used by Gilad et al. (2020) and Muzzarelli et al. (2019). In Muzzarelli et al. (2019), the authors performed 100 runs, initializing 20 times in each run before choosing the best model as defined by reconstruction error. In Gilad et al. (2020), the authors initialize a large number of times, optimizing for hundreds of iterations, and chose the best model defined by reconstruction error to continue. In order to perform this, we would need to run 2000 times for the Muzzarelli et al. (2019) method and 10000 times for the Gilad et al. (2020) method.[2] In addition to the considerable length of time required, we do not believe it is fair to arbitrarily run one method an order of magnitude more times than another. Instead, we focus on running all methods for the same number of initializations. For methods that rely on differences across random initializations, this means that they do not get the added benefit of testing against thousands of initializations.

In order to enforce these requirements, a tolerance may be provided to the package by Lin & Boutros (2020). For the package by Gilad et al. (2020), minor modifications to the code are required. The code is modified to force running all 100 iterations and the `init_factor_matrices` function is modified to return the first generated $W$ and $H$.

## 6 Results

In this section, we present results and compare against other methods. In our analysis of the PBMC3k, NewsGroup4000, and Web of Science datasets, significant computational hurdles were encountered. Due to RAM constraints on our system, we are forced to run CV2k with only 16 threads for both the PBMC3k and NewsGroup4000 datasets. This limitation results in an excessively prolonged runtime of over 30 days for PBMC3k and longer for NewsGroup4000. Given the size of the Web of Science dataset, a further reduction in threads would be necessary, leading to an even longer computational burden.

NNLM-CV is not constrained by memory requirements but is unable to complete even a single run over all ranks on the PBMC3k dataset in 24 hours, meaning the full 100 runs would take over 100 days. The larger scales of NewsGroup4000 and Web of Science implies that completion times would be significantly longer. This difficulty, owing to the incredible time complexity of the method as described in Equation 4, lead us to the conclusion that the burden of computation for these methods is infeasible.

Consequently, the result table for these methods on the mentioned datasets are denoted as "N/A". In other instances, where the output was inconclusive, we have designated the results "undetermined" (Und.). For instance, undetermined is noted when using the elbow method and there is no clear elbow or when using

---

[2]A number of initialization was not described in the CV2K paper but was instead found on the Github linked in the paper.

the permutation method but the slope of the permuted is steeper than the slope of the unpermuted data from the beginning. The remaining results are presented either as integer values or in an increasing order set format, which is applicable only to permutation as well as MCI-RSIC as previously described.

## 6.1 Single Cell Datasets

The resulting suggested ranks for each of the single-cell datasets are shown in Table 1. We show the RSIC-MCI metric as a function of rank on the ALL-AML dataset in Figure 2. The horizontal axis, representing rank, ranges from 2 to 38. Based on our selection criteria described before, we find rank 5 to be the first island of stability. Note that ranks 2 through 5 all have the same or similar MCI values, but rank 5 is the first rank before a significant increase in MCI.

We find that our method and permutation perform poorly on the ALL-AML dataset, whereas cophenetic, dispersion, and CV2k all converge on the generally agreed upon rank and ARI and KS-CV converge on the number of underlying classes. For PBMC3k, no method returned a solution that contended with the number of underlying classes, although MCI-RSIC and elbow both return nearby ranks.

---

[3]Recall that this dataset has no factual underlying classes.

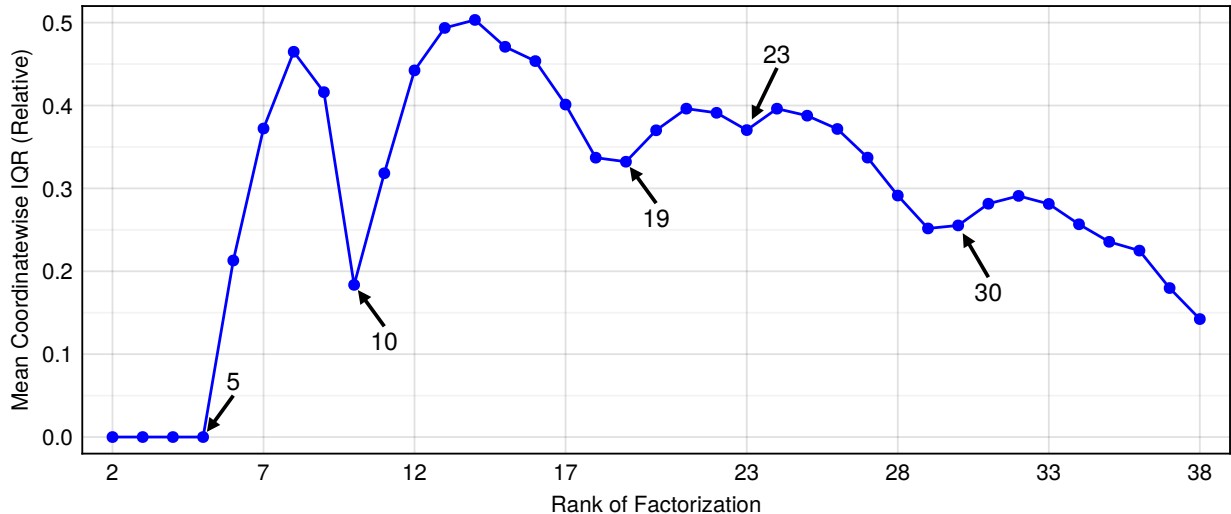

Figure 2: Mean Coordinatewise IQR (MCI) ($y$-axis) vs rank ($x$-axis) for the ALL-AML dataset for ranks 2 through 38. We identify ranks 5, 10, 19, 23, and 30 as "islands of stability" and thereby potential ranks of interest.

Table 1: Results for all evaluated rank determination methods in comparison to the number of underlying classes on the ALL-AML and PBMC3K single cell datasets.

| Method | ALL-AML | PBMC3k |
|---|---|---|
| MCI-RSIC | {5, 10, 19, 23, 30} | {3, 7, 11} |
| Elbow | 4 | 8 |
| Cophenetic | 3 | 2 |
| Dispersion | 3 | 2 |
| Permutation | 5 | 41 |
| ARI | 2 | 2 |
| KS-CV | 2 | N/A |
| CV2K | 3 | N/A |
| MADImput | 4 | N/A |
| Number of Underlying Classes | 2 | $9^3$ |

## 6.2 Image-Based Datasets

We show the RSIC-MCI metric as a function of rank on the Full Digits dataset in Figure 3. This image plots the MCI metric from rank 2 to 64. Like the other methods tested, this method performs equivalently poorly on the Full Digits dataset when compared to the number of underlying classes. There is significant mixing between the ranks, which is indicative of poor clustering behavior.

In addition to the RSIC-MCI metric on Full Digits, we show the metric on Dig0246 in Figure 4. This image plots the RSIC-MCI metric from rank 2 to 64. This clearly shows an island of stability at rank 4 and provides evidence for one at rank 6.

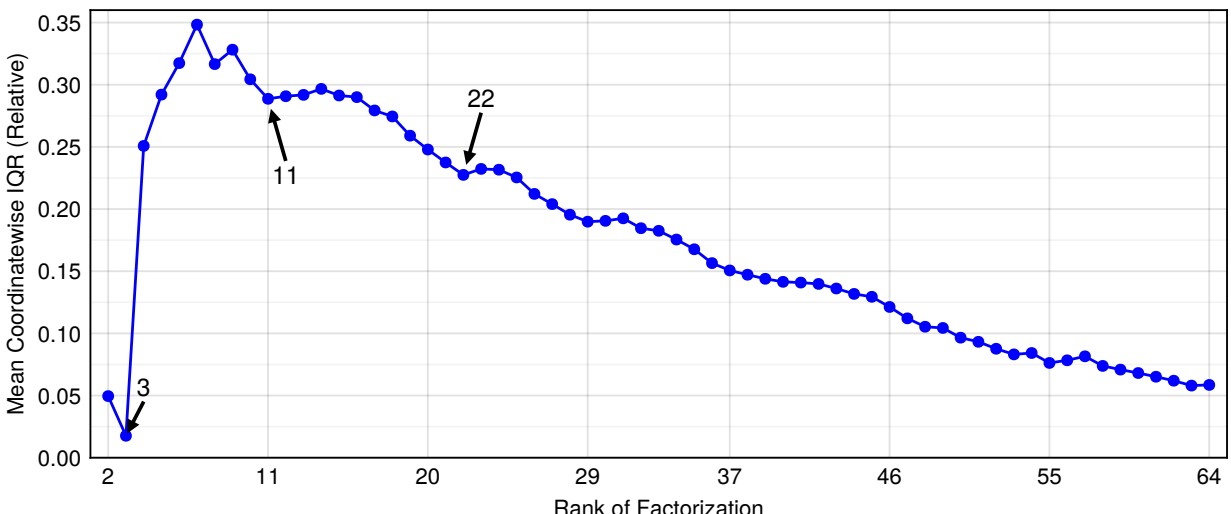

Figure 3: Mean Coordinatewise IQR (MCI) ($y$-axis) vs rank ($x$-axis) for the Full Digits dataset for ranks 2 through 64. We identify ranks 3, 11, and 22 as "islands of stability" and thereby potential ranks of interest.

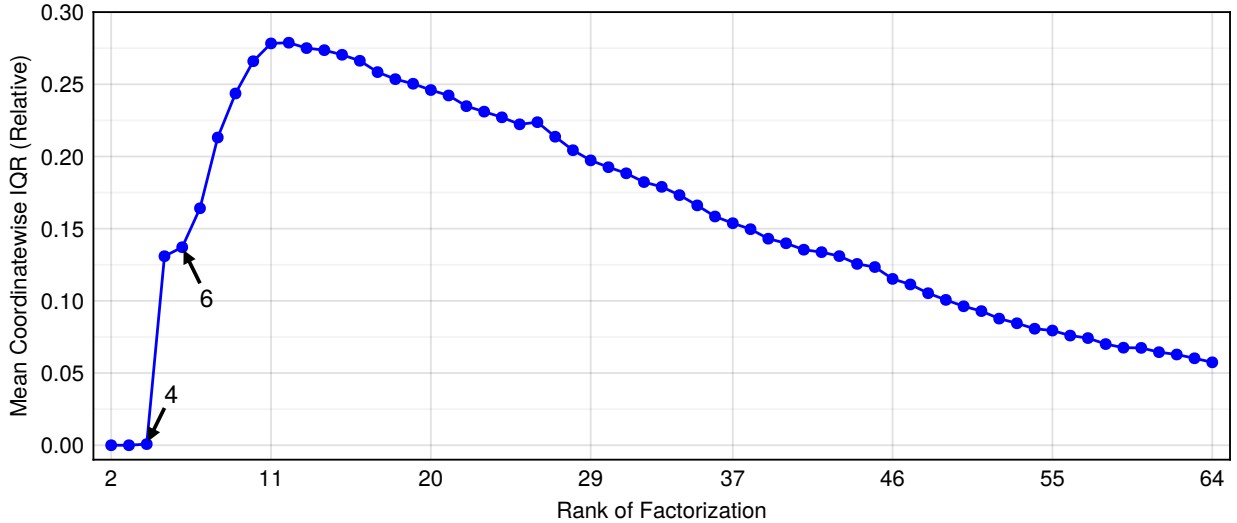

Figure 4: Mean Coordinatewise IQR (MCI) ($y$-axis) vs rank ($x$-axis) for the Dig0246 dataset for ranks 2 through 64. We identify ranks 4 and 6 as "islands of stability" and thereby potential ranks of interest.

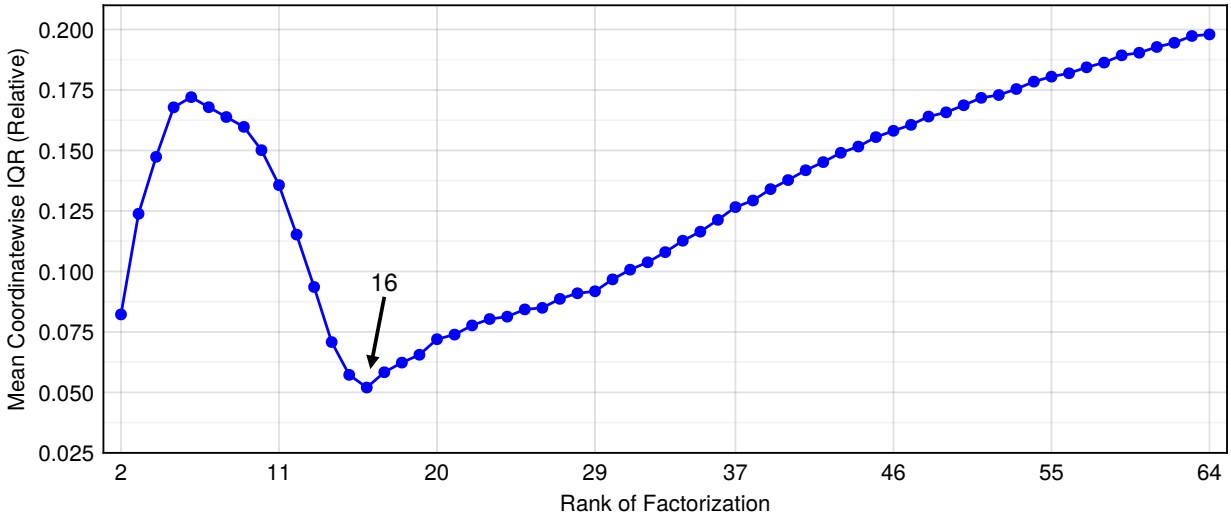

Figure 5: Mean Coordinatewise IQR (MCI) ($y$-axis) vs rank ($x$-axis) for the Swimmer dataset for ranks 2 through 64. We identify rank 16 as the only "island of stability" and thereby potential rank of interest.

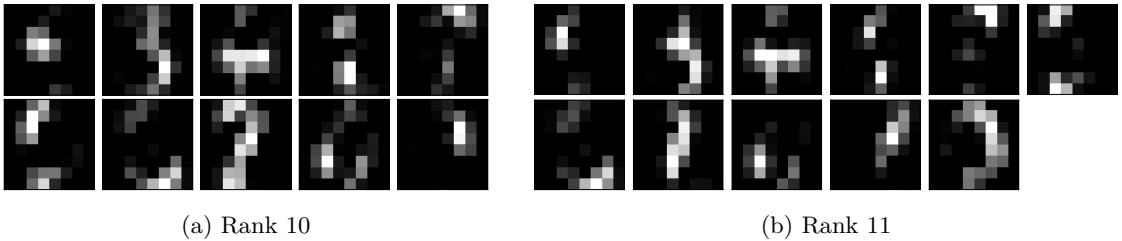

(a) Rank 10          (b) Rank 11

Figure 6: Clustering behavior of the Full Digits dataset at ranks 10 and 11.

Finally, we show the RSIC-MCI metric as a function of rank on the Swimmer dataset in Figure 5. This image plots the MCI metric from rank 2 to 64 and clearly shows an island of stability at rank 16, which corresponds to the number of underlying classes in the dataset.

Figure 6 shows poor clustering behavior at the rank of 10 (number of underlying classes) on the Full Digits dataset. Additionally, we show the clustering behavior at rank 11, which is a nearby rank that we suggest. This figure shows that the clustering behavior at both ranks is rather poor, and provides evidence for why all methods perform poorly on this dataset. Indeed, there appears to be no clear separation between the classes in the dataset. This poor clustering behavior is present in each tested rank, and is indicative of the poor performance of all methods on this dataset.

For the image based datasets, MCI-RSIC and elbow return the true number of underlying classes on the Swimmer dataset. The permutation method consistently detects a rank greater than the number of underlying classes, across all image datasets. For the faces dataset, MCI-RSIC identifies ranks fundamentally different than the number of underlying classes and only elbow chooses a rank matching the number of underlying classes. On Full Digits, all methods perform poorly though the clustering behavior of this dataset is poor as seen in Figure 6. For Dig0246, MCI-RSIC, elbow, cophenetic, dispersion, and ARI all identify a rank matching the number of underlying classes. We note that MADImput found the underlying number of classes in their paper and the discrepancy here is most likely due to their use of 1900 more initializations than allowed in our study (Muzzarelli et al., 2019). The results are shown in Table 2.

Table 2: Results for all evaluated rank determination methods in comparison to number of underlying classes on Swimmer, Faces, Full Digits, and Dig0246 image datasets.

| Method | Swimmer | Faces | Full Digits | Dig0246 |
|---|---|---|---|---|
| MCI-RSIC | 16 | {2, 6, 9, 18} | {3, 11, 22} | {4, 6} |
| Elbow | 16 | 10 | 5 (Und.) | 4 |
| Cophenetic | Und. | 2 | 2 | 4 |
| Dispersion | 64+ | 2 | 2 | 4 |
| Permutation | 18 | 50 | 22 | 17 |
| ARI | 13 | 2 | 5 | 4 |
| KS-CV | 14 | 51 | 12 | 12 |
| CV2K | 17 | 64 | 16 | 16 |
| MADImput | 13 | 60 | 2 | 11 |
| Number of Underlying Classes | 16 | 10 | 10 | 4 |

Table 3: Results for all evaluated rank determination methods in comparison to the number of underlying classes on NewsGroup4000 and Web of Science text datasets.

| Method | NewsGroup4000 | Web of Science |
|---|---|---|
| MCI-RSIC | {4, 12, 24, 46} | {3, 7, 11, 20, 39} |
| Elbow | 9 | 6 |
| Cophenetic | 5 | 3 |
| Dispersion | 12 | 3 |
| Permutation | Und. | 64+ |
| ARI | 3 | 3 |
| KS-CV | N/A | N/A |
| CV2K | N/A | N/A |
| MADImput | N/A | N/A |
| Number of Underlying Classes | 20 | {35, 7} |

### 6.3 Text-Based Datasets

No method selects a rank matching the true number of underlying classes for NewsGroup4000, although MCI-RSIC and dispersion both return a nearby rank. The others underestimate more substantially (elbow, cophenetic, dispersion, ARI), are infeasible to run (KS-CV, CV2K, MADImput), or failed to identify any rank (permutation).

For Web of Science, MCI-RSIC identifies the number of underlying classes for the category as a rank of interest and a nearby rank to the number of subcategories. Elbow detects a nearby rank for the category. All other methods either substantially undershoot (cophenetic, dipsersion, ARI), are infeasible to run (KS-CV, CV2K, MADImput), or indicated the rank was greater than the tested range (permutation). These results are shown in Table 3.

## 7 Discussion & Conclusion

In this paper, we introduced RSIC, a novel method for determining ranks of interest in NMF. Unlike traditional methods which aim to identify a single optimal rank—often requiring extensive parameter tuning and domain-specific knowledge—our approach identifies multiple, possibly relevant ranks by analyzing the sensitivity of the reconstruction residual to different initial conditions (random initializations in the case of this paper). This allows for a more nuanced understanding of the data's underlying structure and provides some flexibility in exploratory data analysis.

Our method identifies "islands of stability", which are ranks where the NMF solutions are less sensitive to initialization and, therefore, more likely to represent meaningful decompositions of the data. We quantified

this stability using the Mean Coordinatewise Interquartile Range (MCI) of the relative reconstruction error across multiple initializations. By doing so, we highlighted ranks that consistently produce stable and interpretable factors, providing insights that single-rank methods may overlook.

We evaluated RSIC on a diverse set of datasets across various domains, including single-cell gene expression data, image datasets, and text corpora. Our experiments demonstrated that RSIC effectively identifies ranks of interest that are consistent or close to the number of underlying classes in the data. Of note, our method performed well on large-scale datasets where other methods tended to undershoot or where cross-validation-based approaches were infeasible due to their high computational complexity.

Comparative analysis with existing methods, including consensus-matrix methods like cophenetic correlation coefficient and dispersion coefficient, self-comparison methods like the ARI, and cross-validation approaches, showed that RSIC is competitive and often superior in identifying meaningful ranks.

However, our method is not without limitations. In datasets where the underlying structure is less pronounced or when the data does not exhibit clear stability islands, RSIC may suggest multiple ranks, requiring further analysis to select the most appropriate one. For example, in the ALL-AML dataset, RSIC showed perfect stability for all ranks at or below 5, but we select 5 based on our selection criteria—this is a limitation of our method. Additionally, while our approach reduces the computational burden compared to some methods, it still requires multiple NMF computations across a range of ranks and initializations.

For future work, we plan to refine the RSIC metric to better handle datasets with subtle or hierarchical structures. Incorporating additional criteria, such as sparsity constraints or domain-specific knowledge, may help to further refine the ranks suggested by RSIC. Additionally, we believe that RSIC could benefit from other types of initialization schemes, such as those based on clustering, dimensionality reduction techniques, or other randomization schemes, to further explore the space of possible initializations. Further, the method could benefit from a window-based smoothing (e.g., Savitzky-Golay) of the MCI values to reduce the noise in the output, which could be particularly useful in datasets with high variability in the reconstruction error. We also aim to explore the theoretical underpinnings of the observed islands of stability to provide deeper insights into why certain ranks yield more stable decompositions.

RSIC offers a robust and significantly more scalable approach for rank suggestion in NMF, taking steps toward bridging the gap between the need for meaningful data decompositions and the practical constraints of computational resources. By providing a selection of relevant ranks and highlighting areas of stability, our method empowers practitioners to make more informed decision in exploratory data analysis and dimensionality reduction tasks.

**Broader Impact Statement**

The proposed RSIC method enhances the applicability of NMF in various fields by provide a more reliable and computationally efficient means of determining relevant ranks. This can benefit areas like bioinformatics, image processing, and text mining, where understanding the underlying data structure is crucial. However, as with any data analysis tool, care must be taken into considering when drawing interpretations drawn from NMF decompositions are valid and do not inadvertently reinforce biases present in the data or researchers.

**Author Contributions**

Both **Marc Tunnell** and **Erin Carrier** jointly contributed to the conceptualization of the project and different methods used in the rank suggestion method and both were involved in conceptualization of the experiments and analysis of the results. **Marc Tunnell** implemented the method, implemented the experiments, and drafted the manuscripts. **Zachary J. DeBruine** provided insight into NMF and cross-validation and was crucial in funding acquisition. **Erin Carrier** provided guidance on the methodological approach, oversaw the project, edited the paper for clarity and coherence.

**Acknowledgments**

This project was in part funded by the Chan Zuckerberg Initiative Data Insights Grant DAF2022-249404. This research made use of computational resources in the Distributed Execution Network (DEN) lab at GVSU.

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
