# OpenReview forum: "Rank Suggestion in Non-negative Matrix Factorization: Residual Sensitivity to Initial Conditions (RSIC)"
_TMLR — Accepted by TMLR_

### Review · Reviewer_iw4J · 2024-11-30

**Summary Of Contributions:**

The paper argues that non-negative matrix factorization is unstable due to its random initialization + optimization nature. Moreover, the authors argue that there could exist several "good" ranks for a given dataset, and the "goodness" is reflected by the spread of solution residuals under a certain rank with random initializations. To this end, the authors proposed to find the "good" ranks with MCI metric over multiple random initializations, where lower MCI metric indicates the optimal NMF solutions are less unstable. The authors provides several simulations to demonstrate how to find the good ranks for different datasets.

**Audience:**

Yes

**Claims And Evidence:**

No

**Requested Changes:**

Please see weakness.

Minor:

1. Figure 1 is not so informative as it just seems errors are scattered throughout the datasets, which is a common phenomenon. The authors could explain it more in detail, preferably with some mathematical language.

2. There are some small typos throughout the paper, e.g., in Sec 2.2, though it ("is" missing) largely domain-dependent, the footnotes 3 in Table 1 points to nowhere.

**Strengths And Weaknesses:**

Strength:

1. The paper investigates an interesting problem, that is exploring the relationship between the stability of non-negative matrix factorization and the target rank.
2. The paper proposes a new metric, MCI, that could potentially capture multiple ranks with relatively good statistical stability in NMF.
3. The paper conducts a large set of numerical simulations to demonstrate how to get the potential "good" ranks in NMF with MCI metric.

Weakness:

1. The proposed solution, identify "island of stability" using the MCI metric, is not rigorously defined. To my best knowledge, the criteria of finding the "island of stability" is only verbally explained without any rigorous mathematical definition in Sec 3. Moreover, the verbal explanation is "island of stability" exhibits "diminished variance" or "minimal compared to nearby ranks", however, there is no clear definition of how much decrease is diminishing, and how many neighbour ranks count as nearby. Later on, the criteria seem to change to the point before large spread increase. The criteria need to be carefully defined and discussed.

2. To identify "good" ranks, the authors are finding ranks that follows the vaguely defined definition. However, note that the MCI metric is an empirical average of residuals. Therefore, error bars should be provided in Figure 2-5 to convince readers that the identified ranks are not simply due to numerical errors.

3. The authors did not provide any explanation behind the "goodness" of ranks besides the simple argument of the more stable the better. This is partly reflected by that fact that the "good" ranks identified by MCI metric are usually not the same as the "true rank".

Overall: The paper, at this moment, serves more as a vaguely-written manual for a new approach in identifying stable ranks in NMF, rather than a scientific paper that provides empirical and theoretical justification for the proposed approach. Although the quantity of numerical simulations is large, the quality of the simulations needs improvement. A smaller set of simulations that examines different aspects of the algorithm with more nuanced discussion, in my opinion, is much better than the current simulation section.

---

> ### Author Response · Authors · 2025-01-12
> **Response and Request for Clarification**
>
> Thanks for the detailed review (and apologies for the terse response due to character limits).  NMF rank detection is subjective; there is often no right answer. Ultimately, the evaluation of rank detection happens by subject matter experts determining whether the decompositions at given rank(s) provide meaningful insights. We believe our method addresses some primary shortcomings in other NMF rank determination methods that will provide these users insight into additional, potentially meaningful, ranks.
>
> Weaknesses:
> 1. We recognize your concern; however this is the norm for most NMF rank detection methods. We do not claim our method is fully automated; they rarely are. For example, the common elbow method is based on visual plot inspection. While we could try to identify a mathematical definition, identifying parameters is often finicky and it can be more clear from visual inspection. However, we recognize the concern about the different wordings you mentioned. The diminished variance and islands of stability terms are meant as conceptual explanations. The minimal compared to nearby ranks was meant more precisely. The point before a large increase was meant as a clarification on which rank should be selected when there is a dip with multiple consecutive ranks at the same level. We will work on clarifying in the revised PDF when a wording is describing a conceptual idea, meant as a definition, or a further clarification.
> 2. You indicate "the MCI metric is an empirical average of residuals" and suggest error bars for figs 2-5. Could you clarify? The metric is not simply an empirical average of residuals. It is an average of pointwise IQR of relative residuals, i.e. relative reconstruction error. We use relative reconstruction errors to provide context for the scale and ensure these are not simply numerical errors. While we mention we use relative reconstruction error in text, the math does not address the relative aspect, which we will fix. The IQR is computed over the 100 different random inits. By using the IQR over random inits, the variability of different inits is fundamentally built-in to the metric itself in a way that is not impacted by outliers. We do not see how it would be valid/sensical to add error bars over different random inits when we are already presenting the IQR. Is your concern with the mean computed over all of the different points (e.g. how much the IQR varies across different points)? We intentionally chose mean over median as the IQR removes outlier inits, but points with an outlier IQR indicate substantial instability. Can you provide clarification so we can address this concern?
> 3. This is tied to the subjective nature of NMF ranks. We see your concern; "good" ranks may not have been the best terminology.  Unfortunately there is no one best rank and how good any given rank is depends on how useful it is to applications specialists (who may find relevance in a variety of different ranks). A more accurate terminology may be "potential ranks of interest". Ultimately, the usefulness of the ranks depends on what subject matter experts are able to draw from that rank decomposition. Also, we thought we made our use of "true rank" clear. There is no true rank; we use "true rank" for the number of underlying classes, but as we attempted to describe in the paper, NMF is not inherently a classification technique. We primarily chose this definition for consistency with the literature. Would you find "number of underlying classes" clearer?
>
> Minor comments:
> 1. Our intent was not simply to show reconstruction errors are scattered throughout a dataset. This figure does not simply show pointwise reconstruction error; it shows the delta between the smallest and largest reconstruction errors (over different random inits) at each point for roughly equally good models. The point was to show *how they are scattered throughout the dataset in very different ways depending on the random initialization*. Was this unclear? If so, we are happy to revise the text and caption. Or, do you still feel this was obvious? As the conceptual basis of our method is that some ranks are inherently more unstable than others, we felt it relevant to visually demonstrate how much variance can exist across similarly good random inits. We are also unsure what you mean by more mathematical language -- could you provide further clarification on what part of the following wording "This figure shows the delta between the smallest and largest error at each point in the rank-10 reconstruction of the first face in the Faces dataset (described in detail later in subsubsection 4.2.3). The maximum delta figure is plotted over the models which had within 10% Frobenius norm of the residual of the median residual model." you did not feel was sufficiently mathematically described? We are willing to clarify the wording, but would appreciate further clarification.
> 2. We fixed these typos and will upload the revised PDF once all reviews are in.

---

> > ### Comment · Reviewer_iw4J · 2025-03-03
> >
> > Thank you for your response. However, my concerns are still large unresolved.
> > Weakness:
> > 1. My concern is not only regarding the wording. In fact, the paper proposes a metric that is used to capture "good ranks", however, there is a lack of rigor in definition and even inconsistency when using this metric (One such example is in Figure 1, in the two rank pairs (18,19) and (29,30), the authors suggested 19, 30 as the "good ranks", which I do not understand the reasoning behind it, why not 18,29 or 19,30?), makes it difficult for researchers or practitioners to appreciate the metric.
> > 2. NMF solution is not unique due to non-convexity, therefore, the IQR of relative reconstruction error is still a random variable subject to the random initialization of W and H matrix. Hence the concern, the "good rank" could be chosen out of randomness.
> > Comments:
> > 1. I understand the metric being plotted in Figure 1. The confusion come from the usefulness of plotting the delta. NMF is a non-convex optimization problem, which means under same training loss (roughly equal average reconstruction error), there exist multiple solutions the are expected to behave differently across different data, isn't this self explanatory? Also, upon further examination, there is also an inconsistency between the explanation of Figure 1. Section 3 says Figure 1 is the reconstruction delta figure for the first face, while Figure 1 shows all 400 faces in the dataset.

---

> > > ### Author Response · Authors · 2025-03-07
> > > **Addressing Clarified Questions/Concerns**
> > >
> > > ### 1. Good Ranks
> > >
> > > We thought we had additional clarification in there, but upon further reading
> > > we understand your confusion. What we choose is not inconsisent
> > > (e.g. in the case of multiple similar, we always choose the same one).
> > > But, we can see how what specifically is chosen was not clear.
> > > We've added additional wording in section 3.2 to address this.
> > >
> > > We would also like to point out that our choice to of how to use
> > > the metric was done prior to examining the results applied to
> > > the datasets.  The choice to choose the largest before a jump in the case
> > > of flat-lining was based on the logic that you effectively get further
> > > decomposition that is equally as stable.  However, it could
> > > be beneficial for a practitioner to exam all of those ranks.
> > >
> > > ### 2. Error Bars / Randomness
> > >
> > > We appreciate this suggestion.
> > > We still want to emphasize that the mean is not across the random initializations,
> > > but instead across the individual points in the reconstruction.
> > > Across the random initializations, we compute the coordinatewise IQR,
> > > which was specifically chosen as it is a metric which measures variability,
> > > but is not affected by outliers.
> > >
> > > The point of the mean is to capture a general picture of this variability at all points.
> > > So, while it is affected by the random initializations, the fact that it is across coordinates
> > > means it measures variation across random initializations varies across points.
> > > While we could put error bars, our concern is that they will be
> > > misinterpreted given the mean is computed across the coordinates, not across random
> > > initializations.
> > >
> > > In light of your concerns, however, we did look at the distributions across
> > > all coordinates for each rank.  These graphs are a bit messy as they are 3D graphs,
> > > but what we see is that the errors tend to be primarily concetrated at the low end,
> > > though there are some (albeit a very tiny percentage) that are substantially larger.
> > >
> > > We also looked at displaying the IQR across the *points* in addition to the mean
> > > (e.g. displaying both the IQR computed across coordinates of the coordinatewise
> > > IQR in addition to ethe mean computed across coordinates of the coordinatewise
> > > IQR).  We find that the general shape of the IQR limits,
> > > both upper and lower, track the shape of the mean.
> > > More specifically, the upper limit (75th percentile), tracks extremely closely
> > > in shape with the mean and the lower limit (25th percentile), is slightly more smoothed,
> > > but still tracks very closely with the general pattern.
> > > This is indicative of capturing what we ended to capture, namely
> > > behavior that is representative of the average amount of variability,
> > > and that the behaivor shown is not simply random noise.
> > >
> > > Note, it is neither the values nor the spread that we care about, but the **shape**.
> > > Because of this we are hesitant to include the visualizations with this displayed
> > > as a reader may incorrectly focus on the values rather than the shape.
> > > The fact that the shape tracks, indicates that the behavior we see is not "random noise",
> > > but instead indicative of relatively common behavior.
> > >
> > >
> > > Note that while the IQR computed coordinatewise across the random initializations
> > > means the variability at each point is not skewed by outlier initializations, as across points
> > > we compute a mean, we are including what one might call "outlier points".
> > > This was intentional -- we didn't want to totally exclude outlier points as
> > > those points are simply the ones for which the reconstructions vary the most
> > > (in a way not influenced by outliers) and is one of the aspects we wanted to capture.
> > >
> > > We also looked at the median, which would not be subject to outlier points,
> > > however we chose the mean as points for which the IQR across initializaitions
> > > is high we felt had some meaning, but should not totally dictate the behaivor.
> > > Much like the lower limit (25th percentile), the median tracks
> > > the same general trend as the mean, albeit slightly smoother than the mean.
> > > If there are deep concerns with the use of the mean, we would be open to replacing it with
> > > median, although due to the slight smoothing, it would mean updating the suggested
> > > ranks  (which would be generally close, but may drop one or two suggested ranks).
> > > We want to stress however that due to the shape of the IQR as described above,
> > > the mean is not simply random noise.
> > >
> > > Would it help if we added a description of some of this behavior to the paper regarding
> > > the distribution of the coordinatewise IQRs in addition to the mean?
> > >
> > > ### 3. Figure 1
> > >
> > > Yes, this is obvious and self-explanatory to us and we understand that this is
> > > likely obvious to readers with a strong background in optimization methods.
> > > However, we also recognize that readers of the paper and users of the metric
> > > are potentially just as likely to be practitioners in a domain discipline,
> > > so we thought it would be helpful and motivating to those readers, who may
> > > lack the formal optimization background.
> > >
> > > With regards to the inconsistency here -- this was simply a typo in the
> > > text which has been fixed.

---

### Review · Reviewer_sp8L · 2024-12-25

**Summary Of Contributions:**

This work proposes a new method for determining ranks in Non-negative Matrix Factorization problems. The main contribution is the proposal of a novel approach (RSIC) that determines a selection of ranks as opposed to one rank. The numerical experiments on a large selection of data suggest that the proposed method's performance is comparable to existing methods.

**Audience:**

Yes

**Claims And Evidence:**

Yes

**Requested Changes:**

1) **Computational Complexity:** This work emphasizes the complexity burden that methods such as Cross Validation present. It also presents RSIC as a "scalable" and "generalizable" method without any study of the complexity of the method. Even a numerical analysis of the complexity (such as comparing computation times with other methods) would be appreciated.

2) **Usefulness of Approach:** It would be beneficial for the author(s) to further discuss why knowing a range of ranks may be useful, or introducing (even briefly) a specific application where one may be more interested in knowing a collection of ranks of interest, or where knowledge of the "island of stability" can be leveraged.

3) **Choosing the Rank:** The question of which rank is ultimately chosen from a collection of computed ranks is, in my view, an important open question and a crucial part of this work. In Table 2, MCI-RSIC performs comparably to Elbow, even when the latter "undershoots". MCI-RSIC provides a collection of ranks but the wrong choice of rank will also lead to undershooting. I appreciate that the author(s) recognized this ad future work but some discussion or insights must also be included in this work.

**Strengths And Weaknesses:**

**Strengths**

The description on previous work on Rank Determination is appreciated, and allows the reader to understand the current research landscape on Rank Determination, as well as the strengths and weaknesses of each method. Additionally, the diverse set of datasets in the numerical analysis is also useful and interesting.

**Weaknesses**

This work seems to me as an interesting proof of concept for a novel Rank Determination method. However, I feel that the main weakness is the completeness of the paper itself. I am unsure if the scope of study is enough for publication as a journal paper. I have written some comments below and hope they are useful.

---

> ### Author Response · Authors · 2025-02-02
> **Response (and thank you for the suggestions)**
>
> Thank you for your detailed, thoughtful review.
>
> With regards to your concern as to whether the scope of study is enough for publication as a journal paper, we do somewhat recognize your concern.  We feel that the conceptual backing, experimental results, and potential benefit is in line with other papers in the field of NMF rank detection.  We believe this metric is of broad interest. We also intentionally submitted to TMLR rather than JMLR, recognizing that the scope of the paper is far more inline with TMLR rather than JMLR, as the former is designed to be more similar to a conference venue whereas the latter is a journal.
>
> With regards to the specific changes you've requested:
> 1. We appreciate this suggestion -- we had previously assumed it was clear.  As it clearly is not, we have added additional details on the complexity of our method / computing the metric.  We will upload the revised PDF once all reviews are in and we have incorporated feedback from all reviewers.
>
> 2. We appreciate this suggestion.  As one example, bioinformatics researchers cannot examine and glean insights from the raw data due to its size.  In these cases, rank determination simply identifies a rank they should look at where they might glean useful information.  These users fully recognize that there is no single rank that matters, and it isn't directly the NMF decomposition they care about but what they can glean that matters.  They simply need to know which rank decompositions to consider.  They are okay with looking at a few different ranks.  We've added an additional discussion of this in the introduction to clarify.  Furthermore, we've also added a subsection to the background and related work section, which elaborates on specifically why this is beneficial for bioinformatics applications and highlights related work that also supports this claim.  As with point 1, we will upload the revised PDF once all reviews are in and we've incorporated feedback from all reviewers.
>
> 3. We believe this is closely tied to point #2.  Ultimately, there are many applications where it is not a question of ultimately choosing a single rank, but instead just identifying which ranks might be useful for researchers.  Additionally, even the concept of "undershooting" is tied to the concept of a "true rank", which we adopt by convention, but may have no practical meaning in the case of an NMF decomposition.  Ultimately, our goal is to identify ranks from which users are potentially more likely to glean useful information.  We believe the discussion in the introduction and in background/related work will likely address this.  If when we upload you believe further discussion would be beneficial, we would be happy to add additional in the conclusion.

---

> > ### Comment · Reviewer_sp8L · 2025-03-02
> > **Response to Authors**
> >
> > I would like to thank the authors for taking the time to respond to my questions. I particularly appreciate the provided example in bioinformatics. I also recognize the section in the revised paper where the complexity of the proposed method is discussed. I believe that discussion will be useful for readers, even practitioners.
> >
> > I do still think that the main limitation of this work is choosing the "best" rank from a collection of ranks. I do recognize that the authors have mentioned this but have yet to provided any meaningful insights into solving this problem. Perhaps it is aimed for future work.

---

### Review · Reviewer_XcKY · 2025-02-03

**Summary Of Contributions:**

Non-negative matrix factorization decomposes a matrix with non-negative elements into a factor and weight matrix, both of which are also non-negative. Like many other unsupervised learning methods such as $k$-means clustering, NMF relies on the specification of a rank $k$, i.e, the latent dimension of the factorization. One well-known example for choosing the rank is the elbow method, where the objective function value is plotted against the rank. The suggested rank is the first rank after which the improvement in terms of objective function value is strongly diminishing.

This paper proposes a new method called Mean Coordinatewise Interquartile Range Residual Sensitivity to Initial Conditions (MCI-RSIC) which aims at suggesting multiple suitable ranks instead of a single rank. The idea is to compute, per considered rank $k$, a number of correlated initializations, compute NMF per initialization, compute the residuals per NMF (per initialization), and finalluy to compute the mean coordinatewise interquartile range. After plotting the proposed metric against the potential ranks $k$, so-called "islands of stability" can be identified where the metric changes.

The authors compare their proposed approach on a number of (classification) datasets against baseline methods for rank suggestion for NMF.

**Audience:**

Yes

**Claims And Evidence:**

No

**Requested Changes:**

Please respond to my questions and weakness points and implement the changes.

**Strengths And Weaknesses:**

### Strengths
1. The paper is easy to read and follow.
2. The paper studies an interesting problem of the matrix factorization community.
3. The proposed approach is novel.


### Weaknesses

1. **The experiments are based on the concept of a "true rank" which I consider problematic.** The authors state in Section 4 that they define the "true rank" of a dataset as the number of classes of the underlying classification dataset and then base their evaluation on that "true rank". I think this is highly problematic. NMF embeds data within a cone and the rank $k$ can be seen as the number of vertices. While it would be nice that every class has its own vertex (corner), this is not necessarily true. Consider handwritten digits. One could hope that 10 is the perfect rank as there are 10 digits, but there are different variants to write digits, thus, one could also argue for a $k$ that is higher than 10. Another example would be where a class is not a single mode but rather multi-modal.
2. **The experiments are not convincing.** Even if we believe that there is a "true rank" which is the number of classes of a classification dataset, I do not consider the experiments convincing. Consider any table. The proposed method MCI-RSIC is not really better than any other baseline. While it sounds nice that the proposed method can suggest multiple ranks, which one should be chosen? I do not often see the "true rank" within the suggestion. Besides, one could also argue that the elbow method is able to suggest multiple ranks.
3. **Time is crucial factor which was not at all evaluated.** Assume a practitioner is interested in a new method for rank suggestion. A critical aspect apart from the performance is time. The proposed method MCI-RSIC relies on many different initializations per potential rank. This is time-intensive. Thus, computation time should be reported. It could be that the suggested method is factors slower than all baselines which often also give reasonable rank estimates. Is the effort worth it? The reader does not know.
4. **The initialization of NMF is very important and it is insufficiently discussed and evaluated.** As mentioned before, many different initializations per potential rank. It is known that the performance of NMF (like many other methods) heavily relies on the initialization. I am missing a discussion of the state of the art when it comes to initialization a NMF model. Would a better initialization or a more diverse set of initializations help? It remains unclear.
5. **An ablation study is missing.** Continuing the previous point regarding the initializations (would initializations other than uniform be better?), it would be interesting to see whether 100 initializations are really necessary. It is unclear how much the rank suggestion relies on the number of initializations.
6. **The concept of "islands of stability" is weird.** In Figure 2, 29 (and 38) is no "island of stability" although 19 and 23 are. Why? Because the bump is not as big? But then in Figure 3, 21 is an "island of stability".
7. **The order of presentation is not always ideal.** Experiment-specific choices should reside in the experiment section but are presented throughout the paper, see Section 2.1 (how to optimize NMF), Section 2.3.3 (evaluation criteria), Section 2.3.4, and Section 3.


### Questions
1. What are the rows and columns in Figure 1? What is the range of the axes (0-1200)? If just the first 400 images are meant, please state 400 in the caption and drop the axes ranges.
2. I have a problem with the statement "We define the 'true rank' of a dataset as the number of classes in the underlying dataset." Please see my weakness above. One can easily define new datasets where the number of dimensions, classes, and optimal rank is arbitrary, meaning the number of classes is different from the optimal rank. Why should the number of classes be a good indicator for the "true rank"?
3. If you force the optimization to run 100 iterations, why do you also set a tolerance? Either you always ran 100 iterations or you stopped when the tolerance criteria is met, but not both. Can you clarify?
4. How can I interpret the seed 123456789? Was it set once before the 100 initializations? You could have also set a seed of $r$ for the $r$-th initialization.
5. In Section 6, RAM constraints were mentioned. I wonder how much RAM there was. Runtimes of 30 days seem unrealistically long for such small problems. Is a single run meant, meaning one optimization for one initialization?
6. Figure 2: If ranks 1-5 all have similar MCI values, why isn't rank 1 preferred over rank 5? Isn't the simpler model always better?
7. Considering the result tables, e.g., Table 1, why should the reader choose the proposed approach given that the true rank is 2, all baselines suggest 2-5 and MCI greater than 5? The elbow seems to do the best job here.
8. In Figure 2, why are 29 and 38 not considered as islands of stability?


### Minor Comments
- Abstract: Typo in "Intial"
- Abstract: Why is Mean Coordinatewise Interquartile Range abbreviated with MCI and not MCIR?
- Abstract: Why is there "existing" in "current state-of-the-art existing rank", if it is the current SOTA, it has exist.
- Introduction, first paragraph: The phrase "complex datasets" can be misleading (datasets consisting of complex numbers).
- Introduction, first paragraph: What are examples for "considerable limitations"?
- Page 2, first paragraph: The references for "Other methods" are missing.
- Page 2, first paragraph (occurs more often): cross-validation-based
- Page 2, first paragraph: It sounds like the preference for lower ranks is a bad thing or not intentional, but in doubt we always prefer the simpler thing. Think about Occam's razor.
- Page 2, second paragraph (occurs more often): Please remove "given" and capitalize section when referring to a specific one.
- Section 2.1: "Although a variety methods" is missing a word.
- Section 2.1: Arbitrary words are capitalized. Why?
- Section 2.1: Space before full-stop: "in this study . Additionally"
- Section 2.1: Two objectives are mentioned but then you choose a third. This is weird. The Frobenius norm is not a single Euclidean distance as suggested in the text.
- Section 2.1: SCD updates: the equations only work after initialization
- Section 2.1: The clustering aspect of NMF drops out of no-where. This should be rephrased.
- Section 2.2: "There are typically numerous equally as good solutions (e.g. factorizations for which the Frobenius norm of the residual are equally small)". Remove "as", use "norms" and "residuals", the norms could also be equally "big"!
- Section 2.2: Full-stop missing: "2008) Third,"
- Section 2.3.1: "unless choosing to run on a limited range of ranks" The range of ranks is always limited!
- Section 2.3.2: "randomly in half"
- Section 2.3.2: Why is Rand capitalized (while adjusted and index are not). Same with AIC in Section 2.3.3.
- Section 2.3.3: Wrong citation style for Yamauchi et al. (2012), and Tan & Fevotte (2009).
- Section 2.3.4: References are missing (MADImput, etc).
- Sections 2.3.4, 4.1, 5.3, 5.5: Many wrong citation styles.
- Section 2.4: The meaning of $\sim$ should be introduced earlier (at least one sentence earlier).
- Section 2.4: Equation (3), please use \eqref
- Page 7, last equation: Please use \text{opt} and \text{init} as before. If opt refers to optimized, I suggest to use NMF, since opt could be interpreted as optimal, which is not true!
- Section 3.2: Please use \cdot for $\text{vec}(\cdot)$.
- Section 3.2: Rephrase "may now"
- Section 4: If a dataset is taken from scikit-learn, please name scikit-learn and cite the paper once. The reader might be familiar with scikit-learn but less so with the last name of the first author of this publication.
- Section 5: There are a lot of tense mismatches. Either, experiments happen as the reader reads the manuscript, or they all happened in the past, but the tenses are mixed.
- Page 10, first paragraph: The Frobenius norm is not defined in Equation (1)!
- Page 10, first paragraph: How much memory and cores does the other machine have? The information is insufficient.
- Section 5.3: "the the"
- Section 5.3: State numpy. However, this sentence (package for shuffling) can also be deleted as it has no useful information.
- Section 5.3: Why is now the median computed when before, the average was computed (Section 5.1)?
- Section 6: A result table is mentioned but it is unclear which one. So far, no table appeared.
- Section 6: I do not understand why Section 5 states that $k_{\text{min}}=2$ but in Section 6 it changes to $k_{\text{min}}=1$ and the figures start from zero. The figures should start from $k_{\text{min}}$ and end with $k_{\text{max}}$. Axis labels should be provided (not only in the caption) and a light grid would make it easier to read.
- Figure 6 is way too big.

---

> ### Author Response · Authors · 2025-02-07
> **Response to Weaknesses**
>
> We split the response for different pieces across different comments due to character limits.
>
> 1. The experiments are based on the concept of a "true rank" which I consider problematic. The authors state in Section 4 that they define the "true rank" of a dataset as the number of classes of the underlying classification dataset and then base their evaluation on that "true rank". I think this is highly problematic...
>     * We understand your concern. We chose this terminology purely for consistency with the literature, but in light of it being a concern for all reviewers, we've opted to rephrase it as "number of underlying classes" to avoid the confusion of "true rank". However, we leave it as a comparison because, although flawed, it is the norm.
>
> 2. The experiments are not convincing. Even if we believe that there is a "true rank" which is the number of classes of a classification dataset, I do not consider the experiments convincing...
>     * We believe this is tied to the use of the terminology "true rank", which we have adapted as described above. Based on convention, we felt potential readers would want to see this comparson. With regards to multiple ranks, in response to another reviewer, we have added substantial background on where multiple ranks, rather than a single rank, may be useful. As the benefit of any given NMF decomposition is subjective, depending on what subject matter experts are able to draw from that rank decomposition. Ultimately, we believe the method seems to be detecting reasonable ranks and noticeable behavior, though confirmation by subject matter experts will be the ultimate evaluation.
>
> 3. Time is crucial factor which was not at all evaluated. Assume a practitioner...
>      * We have substantially expanded the computational complexity. As execution time is fairly arbitrary (and could be improved by switching languages) and relies on implementation details, rather than the fundamental complexity, we feel it is not an accurate representation.  Instead, we focus on computational complexity which is language agnostic.
>
> 4. The initialization of NMF is very important and it is insufficiently discussed and evaluated. As mentioned before, many different initializations per potential rank....
>     * We have expanded our discussion on NMF initializations in the background section. Many "state of the art" NMF initializations are costly or rely on having other factorizations previously computed, while others provide little benefit beyond random initializations. While an expanded upon study across all different types of initializations would be of potential interest, out of fairness and a desire to keep this paper to a reasonable length, we believe it is outside of the scope of this work.
>
> 6. An ablation study is missing. Continuing the previous point regarding the initializations (would initializations other than uniform be better?), it would be interesting to see whether 100 initializations are really necessary. It is unclear how much the rank suggestion relies on the number of initializations.
>     * While we did not perform an official study and doing so is practically infeasible for the scope of this work, we tested a variety of different numbers of random initializations both smaller and larger than 100.  We did not try to pinpoint the exact location, but we did find that 50 was not enough for results to be consistent, whereas 100 provided consistent behavior.  More than 100 (e.g. 200, 500, 1000), did not produce meaningful differences with regards to our metric.  Based on our testing, 100 was a reasonably accurate approximate sweet spot.  We added a sentence to this regard in section 3.
>
> 8. The concept of "islands of stability" is weird. In Figure 2, 29 (and 38) is no "island of stability" although 19 and 23 are. Why? Because the bump is not as big? But then in Figure 3, 21 is an "island of stability".
>     * See answer to question 8.
>
> 9. The order of presentation is not always ideal. Experiment-specific choices should reside in the experiment section but are presented throughout the paper, see Section 2.1 (how to optimize NMF), Section 2.3.3 (evaluation criteria), Section 2.3.4, and Section 3.
>     * We see and understand this point.  However, on the flip side, many of these statements were in the locations that we presented them in order to explain why we don't spend substantial space talking about other, not relevant metrics / objective functions and to keep these sections brief and focused on the points that are actually relevant background.  We think a reasonable compromise might be to leave them where they are, but reemphasize briefly in experimental setup that we limit our experimentation to a subset of related metrics for the reasons described in Background & Related Work.

---

> ### Author Response · Authors · 2025-02-07
> **Response to Questions**
>
> Again, due to character limits we break up the response.  This response is solely to your questions.
>
> 1. What are the rows and columns in Figure 1? What is the range of the axes (0-1200)? If just the first 400 images are meant, please state 400 in the caption and drop the axes ranges.
>     * We fixed the caption and also dropped the ranges.
>
> 2. I have a problem with the statement "We define the 'true rank' of a dataset as the number of classes in the underlying dataset." Please see my weakness above. One can easily define new datasets where the number of dimensions, classes, and optimal rank is arbitrary, meaning the number of classes is different from the optimal rank. Why should the number of classes be a good indicator for the "true rank"?
>     * Frankly, we have a problem with the statement as well but used the term true rank for consistency with some of the literature we compare against. We feel as though we have added discussion that addresses this and have changed the wording regarding true ranks.
>
> 3. If you force the optimization to run 100 iterations, why do you also set a tolerance? Either you always ran 100 iterations or you stopped when the tolerance criteria is met, but not both. Can you clarify?
>     * Some software packages do not have the ability to set a predetermined minimum number of iterations. In order to enforce this requirement, we set an unrealistically low residual value such that it will always go to the maximum number of iterations.
>
> 4. How can I interpret the seed 123456789? Was it set once before the 100 initializations? You could have also set a seed of r for the r-th initialization.
>     * It was set once before the 100 initializations. We had considered doing the suggestion as well but instead just kept track of the current RNG state for checkpointing.
>
> 5. In Section 6, RAM constraints were mentioned. I wonder how much RAM there was. Runtimes of 30 days seem unrealistically long for such small problems. Is a single run meant, meaning one optimization for one initialization?
>     * We update where we state the specifications for the machines. The consumer grade machine has more ram but less cores. Additionally, by one run we mean a run through all ranks for a given initialization or pass. We allowed for 100 initializations, so it would have taken at least another 100 days.
>
> 6. Figure 2: If ranks 1-5 all have similar MCI values, why isn't rank 1 preferred over rank 5? Isn't the simpler model always better?
>     * This was a choice that we made. In practice, a domain scientist may determine that any one of the lower ranks is ideal.
>
> 7. Considering the result tables, e.g., Table 1, why should the reader choose the proposed approach given that the true rank is 2, all baselines suggest 2-5 and MCI greater than 5? The elbow seems to do the best job here.
>      * We did not perform well on the ALL-AML dataset and state such in the text. Choosing the last element prior to a jump in MCI was a choise we made.
>
> 8. In Figure 2, why are 29 and 38 not considered as islands of stability?
>     * We've updated the graph to include 30 (it's actually 30, not 29).  With respect to 38, 38 is the full rank of this dataset.  It's generally very unlikely one would want NMF with k being the full rank, as you no longer have a low-rank decomposition.  More generally though, if there was reason to believe the endpoint of plotted ranks was potentially an "island of stability", one should increase k_max.  However, with the exception of Swimmer (which is an artificial dataset), the long term behavior is that as k increases MCI appears to taper off, so we don't see reason to believe it warrants running for a larger k_max.

---

> ### Author Response · Authors · 2025-02-07
> **Response to Minor Concerns**
>
> * Abstract: Typo "Intial"
>   * Fixed
> * Abstract: Why is Mean Coordinatewise Interquartile Range abbreviated MCI and not MCIR?
>   * The I stands for IQR
> * Abstract: Why is there "existing" in "current state-of-the-art existing rank", if it is the current SOTA, it has exist.
>   * Fixed
> * Introduction, first paragraph: The phrase "complex datasets" can be misleading (datasets consisting of complex numbers).
>   * Fixed, changed to ``hard-to-interpret''
> * Introduction, first paragraph: What are examples for "considerable limitations"?
>   * We swapped that sentence to be more readable. Previously the sentence before it itemized some limitations.
> * Page 2, first paragraph: The references for "Other methods" are missing.
>   * Fixed
> * Page 2, first paragraph (occurs more often): cross-validation-based
>   * Fixed
> * Page 2, first paragraph: It sounds like the preference for lower ranks is a bad thing or not intentional, ...
>   * We believe the new section on multi-rank and hierarchical NMF approaches addresses this question.
> * Page 2, second paragraph (occurs more often): Please remove "given" and capitalize section ...
>   * Fixed
> * Section 2.1: "Although a variety methods" is missing a word.
>   * Fixed
> * Section 2.1: Arbitrary words are capitalized. Why?
>   * Some of these were capitalized due to being names (Rand, Akaike). We have gone back and capitalized the first letter of words in all acronyms, which is what we believe you suggested.
> * Section 2.1: Space before full-stop: "in this study . Additionally"
>   * Fixed
> * Section 2.1: Two objectives are mentioned but then you choose a third. This is weird. The Frobenius norm is not a single Euclidean distance as suggested in the text.
>   * Addressed
> * Section 2.1: SCD updates: the equations only work after initialization
>   * Added a sentence stating we assume appropriately initialized.
> * Section 2.1: The clustering aspect of NMF drops out of no-where. This should be rephrased.
>   * We moved this paragraph to the following section so it follows more naturally.
> * Section 2.2: "There are typically numerous equally as good solutions (e.g. factorizations for which the Frobenius norm of the residual are equally small)". Remove "as", use "norms" and "residuals", the norms could also be equally "big"!
>   * Fixed
> * Section 2.2: Full-stop missing: "2008) Third,"
>   * Fixed
> * Section 2.3.1: "unless choosing to run on a limited range of ranks" The range of ranks is always limited!
>   * This was updated to "relatively few ranks"
> * Section 2.3.2: "randomly in half"
>   * Changed to randomly in half
> * Section 2.3.2: Why is Rand capitalized (while adjusted and index are not). Same with AIC in Section 2.3.3.
>   * Rand was capitalized as it was a name, but we have addressed it as mentioned above
> * Section 2.3.3: Wrong citation style for Yamauchi et al. (2012), and Tan & Fevotte (2009).
>   * Fixed
> * Section 2.3.4: References are missing (MADImput, etc).
>   * fixed
> * Sections 2.3.4, 4.1, 5.3, 5.5: Many wrong citation styles.
>   * Believe we fixed
> * Section 2.4: The meaning of ~ should be introduced earlier (at least one sentence earlier).
>   * Fixed
> * Section 2.4: Equation (3), please use \eqref
>   * We aren't sure what you are saying -- the style is what is produced and matches the style guide.
> * Page 7, last equation: Please use \text{opt} and \text{init} as before. If opt refers to optimized, I suggest to use NMF ...
>   * Addressed
> * Section 3.2: Please use \cdot for vec(*).
>   * Fixed
> * Section 3.2: Rephrase "may now"
>    * Changed to "is computed as"
> * Section 4: If a dataset is taken from scikit-learn, please name scikit-learn and cite the paper once. The reader might be familiar with scikit-learn but less so with the last name ...
>   * We modified the text to refer to scikit-learn and briefly explain what it is on first time mention. We feel it is important to cite the source when mentioned afterward, though.
> * Section 5: There are a lot of tense mismatches. Either, experiments happen as the reader reads the manuscript, or they all happened in the past, but the tenses are mixed.
>   * Fixed
> * Page 10, first paragraph: The Frobenius norm is not defined in Equation (1)!
>   * Fixed
> * Page 10, first paragraph: How much memory and cores does the other machine have? ...
>   * Fixed
> * Section 5.3: "the the"
>   * Fixed
> * Section 5.3: State numpy. However, this sentence (package for shuffling) can also be deleted as it has no useful information.
>   * Removed
> * Section 5.3: Why is now the median computed when before, the average was computed (Section 5.1)?
>   * Median was for consistency with other evaluations. For elbow, there is no discernible diff between median and mean (which we now indicate).
> * Section 6: A result table is mentioned but it is unclear which one. So far, no table appeared.
>   * Fixed
> * Section 6: I do not understand why Section 5 states that $k_{min} = 2$ but in Section 6 it changes to $k_{min} = 1$ and the figures start from zero ...
>   * Fixed
> * Figure 6 is way too big.
>   * Fixed

---

> ### Author Response · Authors · 2025-02-07
> **Thank you for the very detailed review!**
>
> We did not have the space to say this in the other comments due to character limits (and wanting to include enough context where it was clear what we were responding to, but we greatly appreciate your extremely detailed review.

---

> > ### Comment · Reviewer_XcKY · 2025-02-27
> > **Thank you**
> >
> > Thank you very much for your responses.
> >
> > I really appreciate that you answered my questions and incorporated almost all suggestions. It would have been much easier to check the revised version if you had colored the changes. Perhaps something to consider in the future.
> >
> > I was surprised to see an additional term in the equation computing the R matrix in Section 3.2. I believe it was never mentioned in this discussion, suddenly appeared while the results did not change. If it doesn't change the results, why add it?
> >
> > Sadly, some of my biggest points remain somewhat undressed.
> > - An evaluation of computation time. I appreciate the complexity analysis but as a practitioner I would not care much. If I need to initialize 100 times, I have 100 times the complexity of NMF. The added value of this analysis is also negligible.
> > 	- I do not buy the argument of difference of programming languages and way of implementation. If you fix those things across your evaluation, your evaluation is valid. **Ultimately, 100 initializations/NMF-runs are needed which take time.** My point is that this sounds like a slow method and there is no indication whatsoever how the proposed method compares to the state of the art (baselines).
> > 	- Thus, I do not agree with *"The scalability of RSIC is a significant advantage over many existing methods."* as stated in Section 3.3 on page 9. This claim is not verified.
> > 	- Besides, it remains unclear (in the paper) whether 100 initializations are needed and how critical this number is. Note that the number of initializations is a hyperparameter of the proposed method!
> > 	- Considering the tables, the suggested method is not clearly the best (if we believe the story of the true rank) and if it is also considerably slower than the baselines, what is the point of using it?
> > - Apart from the 100 iterations, the need for the correlated initializations is also not evaluated/empirically shown.
> >
> > Not mentioned in my original review but added for completeness:
> > - Any variance measure around the mean coordinatewise IQR could have been shown.
> > - The baseline metrics against the rank $k$ could have also been shown for comparison.

---

> > > ### Author Response · Authors · 2025-03-02
> > > **Further Response / Clarifications**
> > >
> > > Regarding coloring changes
> > > * We appreciate the suggestion and will do so in the future.
> > >
> > > Additional term in the R matrix equation:
> > > * Yes, it was not given in the original manuscript.
> > >   In our revised manuscript, we clarify that the term
> > >   computes the relative residual with the additional
> > >   1 due to the presence of zero values.
> > >   Our use of the relative reconstruction error was
> > >   discussed elsewhere in the paper but was missed
> > >   in this equation. It does not alter the results
> > >   elsewhere as everything was computed with the
> > >   relative reconstruction error.
> > >
> > > Computation time and initialization
> > > * Computation time:
> > >   * It is well-established that the
> > >     choice of programming language can have a large impact
> > >     on the speed of the underlying method.
> > >   * Reimplementing every baseline in a single language or
> > >     reimplementing our method in another language is outside
> > >     of the scope of the paper.
> > >   * Additionally, computation time is ultimately implementation
> > >     dependent, whereas analysis of computational complexity is
> > >     a more meaningful analysis of the amount of work required.
> > >
> > > * Initialization computation time:
> > >   * We recognize the concern regarding the computational
> > >     burden of using 100 initializations. Our choice is
> > >     consistent with standard practice in the literature.
> > >     In fact, all methods we compare against (and effectively
> > >     all baseline methods) require a similar or higher (in one
> > >     case orders of magnitude higher) number of initializations.
> > >   * The computation time to compute 100 initializations is,
> > >     again, the standard accepted practice in the area.
> > >     Our method has a computational cost proportional
> > >     to this, unlike the state-of-the-art we compare
> > >     against and simply could not run due to their
> > >     substantial time requirements.
> > >
> > > * Scalability:
> > >   * As we showed through the complexity analysis, the
> > >     scalability of RSIC as the data grows is as good
> > >     as, if not substantially better than,
> > >     the existing methods.
> > >   * If you disagree, could you be more specific and
> > >     be precise about which specific methods have shown
> > >     through complexity analysis that they have a
> > >     fundamentally better scalability with respect
> > >     to growth in the problem size?
> > >
> > >
> > > Initialization approach:
> > > * Number of initializations:
> > >   * As we stated, while we did not do an absolutely
> > >     comprehensive test to identify the precise number
> > >     of initializations required, 100 is not randomly
> > >     chosen — it was in fact empirically tested
> > >     and found that substantially fewer was not sufficient
> > >     and substantially more provided no significant benefit.
> > >
> > > * Progressive initializations
> > >   * For space purposes, we did not show this,
> > >     but the progressive random initializations are required.
> > >     Without progressive initializations (which we did test),
> > >     the differences in behavior between ranks is not as meaningful
> > >     as it is too subject to noise to be clearly indicative.
> > >   * We thought we made this clear, but if you would like, we can add
> > >     another sentence in commenting on these.
> > >
> > >
> > > Overall Results:
> > > * We don't claim that our method is clearly the best and
> > >   As stated in the conclusion section of our paper:
> > >
> > >   "Our experiments demonstrated that RSIC effectively identifies
> > >   ranks of interest that are consistent or close to the number
> > >   of underlying classes in the data. Of note, our method performed
> > >   well on large-scale datasets where other methods tended to
> > >   undershoot or where cross-validation-based approaches were
> > >   infeasible due to their high computational complexity."
> > >
> > >   NMF ranks are inherently subjective.  How good any given rank
> > >   is depends on how useful it is to applications specialists
> > >   (who may find relevance in a variety of different ranks).
> > >   Ultimately, our goal is to identify ranks from which users
> > >   are potentially more likely to glean useful information.
> > >
> > > * Ultimately, we believe our metric for NMF rank determination
> > >   is of broad interest and feel that the conceptual backing,
> > >   experimental results, and potential benefit is in line with
> > >   other papers in the field of NMF rank detection.
> > >
> > > Plotting Comments:
> > > * Just to be clear, what would this variance be over?
> > >   Given the IQR is already computed across random initializations,
> > >   we fail to see how it would be sensical/valid to further
> > >   compute a variance measure.  Do you mean across all points?
> > >   We had considered metrics other than the mean and found
> > >   the mean provided the most meaningful results.
> > >   We would be open to trying to add this variance measure
> > >   if you find it particularly important, though we are concerned
> > >   that given it is already a mean of an IQR, that the
> > >   meaning may be lost on a reader.
> > > * It is unfeasible to plot the baseline metrics as it would
> > >   require an inordinate amount of space, substantially
> > >   lengthening the paper without adding meaningful information.

---

### Decision · Action_Editor_C6Lv · 2025-03-13

**Recommendation:** Accept as is

**Comment:**

The reviewers and authors engaged in a robust and fruitful discussion. The comments by the reviewers and changes to the manuscript resulted in a much improved submission. There was substantial discussion around the use of evaluative words like "best" and comparison to a "true" rank. These concerns are well founded and the authors made efforts to adjust the claims to rest firmly on the foundation of the evidence. However, the community norms also impinge on the word selection - the authors have little control over those expectations in the short run, though one expects that the field will converge to appropriate descriptions in the long run. Within the bounds of the expectation of the journal the authors have responded to the most important comments. Of course, more work may be done to test the methodology in different ways. The paper seems to meet the standards of the evaluation criteria.

**Audience:**

The paper is appealing to a broad range of the TMLR audience. NMF is a widely deployed matrix decomposition approach in many fields. The authors have attempted to tackle a difficult and contentious problem in an innovative way.

One aspect of the work that I think may not be fully exposited is the connection between these ideas and those found in the area of Veridical Data Science (https://vdsbook.com/). In particular, the idea that conclusions should be _stable_ in response to changes in the pipeline - in this case initializations seems relevant to the results presented in this paper. I look forward to any comments by the authors or community with regard these connections.

**Claims And Evidence:**

This paper approaches the problem of choosing a rank for non-negative matrix factorization with the idea that one should not focus exclusively on a single optimal rank according to one criterium, but instead one might be better served by considering the variability of NMF solution to initial conditions. The authors present theory and empirical evidence to support their approach. The authors have made significant revisions in the manuscript in response to the reviewer comments. With regard to the evaluation criteria of the journal, I believe the conditions for acceptance are met.